# Row-clustering of a Point Process-valued Matrix

**Lihao Yin**[1,2]  **Ganggang Xu**[3]  **Huiyan Sang**[2]  **Yongtao Guan**[3]

[1]Institute of Statistics and Big Data, Renmin University, Beijing, China,
[2]Department of Statistics, Texas A&M University, College Station, Texas,
[3]University of Miami, Coral Gables, Florida,
{lihao, huiyan}@stat.tamu.edu, {gangxu, yguan}@bus.miami.edu

## Abstract

Structured point process data harvested from various platforms poses new challenges to the machine learning community. To cluster repeatedly observed marked point processes, we propose a novel mixture model of multi-level marked point processes for identifying potential heterogeneity in the observed data. Specifically, we study a matrix whose entries are marked log-Gaussian Cox processes and cluster rows of such a matrix. An efficient semi-parametric Expectation-Solution (ES) algorithm combined with functional principal component analysis (FPCA) of point processes is proposed for model estimation. The effectiveness of the proposed framework is demonstrated through simulation studies and real data analyses.

## 1 Introduction

Large-scale, high-resolution, and irregularly scattered event time data has attracted enormous research interest recently in many applications, including medical visiting records [13], financial transaction ledgers [28] and server logs [11]. Given a collection of event time sequences, one research thread is to identify groups displaying similar patterns. In practice, the significance of this task emerges in multifarious scenarios. For example, matching users with similar activity patterns on social media platforms is beneficial to ads recommendations; clustering patients by their visiting records may help predict the course of the disease progression.

Our study is motivated by a dataset we collected from Twitter, which consists of posting times of 500 university official accounts from April 15, to May 14th, 2021. Figure 1 displays posting time stamps of seven selected accounts in five consecutive days. While the daily posting patterns vary across different accounts, the posting date seems to also play an important role. Specifically, all accounts cascade a barrage of postings on April 16th while few postings appear on April 18th. Lastly, each posting is associated with a specific type of activity, namely, tweet, retweet, or reply. Our main interest is to cluster these multi-category, dynamic posting patterns into subgroups.

To characterize the highly complex posting patterns, we propose a mixture model of Multi-level Marked Point Processes (MM-MPP). We assume that the event sequences from each cluster are realizations of a multi-level log-Gaussian Cox process (LGCP) [16], which has been demonstrated useful for modeling repeatedly observed event sequences [28]. We here extend their work to the case of mixture models and propose a semiparametric Expectation-Solution algorithm to learn the underlying cluster structure. The proposed learning algorithm avoids iterative numerical optimizations within each ES step and hence is computationally efficient. In particular, the expectation step is carried out using MCMC samples based on the FPCA of the latent Gaussian processes, which imposes minimal parametric assumptions on the proposed model. Finally, we design an algorithm that can take advantage of array programming and GPU acceleration to further speed up computation.

35th Conference on Neural Information Processing Systems (NeurIPS 2021).

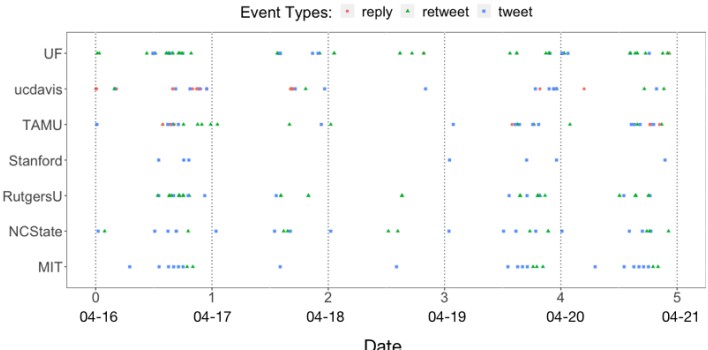

Figure 1: The activities of selected accounts on Twitter.

## 2 Related Work

**Modelling of Event Sequences** Point processes have been widely used to model temporal events [5], although rarely does existing work focus on repeatedly observed event sequences. One prominent example is the Hawkes process [8, 32, 33], which accounts for temporal dependence among events by a self-triggering mechanism. However, the existing Hawkes process may fail to describe our cases for two reasons. First, many human activities naturally have discontinuity by day. So it is unclear how to define the triggering mechanism across days with Hawkes processes. Second, the multivariate Hawkes process characters an overall rate of events for different days. The clustering methods based on the Hawkes process [15, 14, 30] are more likely to distinguish individuals by overall event frequency, other than their intra-day behavior patterns.

In our motivating example, there exist multiple variations for the event sequences, both from individual and day levels. One way to account for variations from multiple sources is to exploit Cox process models, whose intensities are modeled by latent random functions. One popular class of Cox processes is the log-Gaussian Cox process (LGCP) [16], whose latent intensity functions are assumed to be transformed Gaussian processes. Recently, [28] proposed a multi-level LGCP model to account for different sources of variations for repeatedly observed event data. However, clustering of repeatedly observed marked event time data was not considered in their work.

**Clustering of Event Sequences.** Extensive research has been done on this topic. To our knowledge, clustering models for point processes can be summarized into two major categories: distance-based clustering [3, 4, 18] and distribution-based clustering [30, 15]. The former measures the closeness between event sequences based on some extracted features and then uses classical distanced-based clustering algorithms such as $k$-means [4, 19] or EM algorithms [27]. The second approach, also referred to as model-based clustering, assumes that event sequences are derived from a parametric mixture model of point processes. One notable thread is the mixture model of the Hawkes point processes. For example, [30] proposed a Dirichlet mixture of Hawkes processes (DMHP) under the Expectation-Maximization (EM) framework to identify clusters. However, existing EM algorithms for event sequence clustering have a common issue that they typically require iterative numerical optimizations within each M-step, which would drastically overburden the computation. This computational issue will be accentuated when event data are repeatedly observed and have marks.

## 3 Model-based Row-clustering for a Matrix of Marked Point Processes

**Notation.** Suppose that we observe daily event sequences from $n$ accounts during $m$ days. For account $i$ on day $j$, let $N_{i,j}$ denote the total number of events, $t_{i,j,l} \in (0, T]$ denote the $l$-th event time stamp, and $r_{i,j,l} \in \{1, \cdots, R\}$ denote the corresponding event types (marks). The activities of account $i$ on day $j$ can be summarized by a set $S_{i,j} = \{(t_{i,j,l}, r_{i,j,l})\}_{l=1}^{N_{i,j}}$, recording the time stamps and types for all $N_{i,j}$ events. This general notation can also describe other marked event sequences which are repeatedly observed on $m$ non-overlapping time slots. We represent the collection of all marked daily event sequences as an $n \times m$ matrix $\mathcal{S}$, whose $(i, j)$th entry is a marked event sequence

$S_{i,j}$. We aim to cluster the rows of $\mathcal{S}$ to identify potential heterogeneity in account activity patterns, while taking into account the dependence across rows and columns to characterize the complex event patterns and interactions among accounts, days, and event types.

## 3.1 A Mixture of Multi-level Marked LGCP Model

Given a matrix of daily event sequences $\mathcal{S}$, we can separate each matrix entry $S_{i,j}$ according to their marks (event types). Let $S_{i,j}^r = \{t_{i,j,l} | r_{i,j,l} = r\}$ be the collection of time stamps of event type $r \in \{1, \cdots, R\}$. We model each $S_{i,j}^r$ by an inhomogeneous Poisson point process conditional on a latent intensity function $\lambda_{i,j}^r(t|\Lambda_{i,j}^r) = \exp\{\Lambda_{i,j}^r(t)\}$, where $\Lambda_{i,j}^r(t) : [0, T] \mapsto \mathbb{R}$ is the random log intensity function on $[0, T]$. Following [28], we assume a multi-level model for $\Lambda_{i,j}^r(t)$:

$$\Lambda_{i,j}^r(t) = X_i^r(t) + Y_j^r(t) + Z_{i,j}^r(t), \quad t \in [0, T], \tag{1}$$

for $i = 1, \cdots, n$, $j = 1, \cdots, m$ and $r = 1, \cdots, R$. In model (1), $X_i^r(t)$, $Y_j^r(t)$ and $Z_{i,j}^r(t)$ are random functions on $[0, T]$, characterizing the variations of account-level, day-level and the residual deviations, respectively. In addition, we also take into account the dependence across event types when modelling $X_i^r(t)$, $Y_j^r(t)$ and $Z_{i,j}^r(t)$, while assuming independence across accounts, that is, for any $(r, r')$, $X_i^r(t)$ and $X_{i'}^{r'}(t)$ are independent when $i \neq i'$, $Y_j^r(t)$ and $Y_{j'}^{r'}(t)$ are independent when $j \neq j'$, and $Z_{i,j}^r(t)$ and $Z_{i',j'}^{r'}(t)$ are independent if $(i, j) \neq (i', j')$.

We assume that $\mathbf{X}_i(t) = \{X_i^r(t)\}_{r=1}^R$ is a mixture of multivariate Gaussian processes with $C$ components in order to detect heterogeneous clusters. We introduce a binary vector $\boldsymbol{\omega}_i = \{\omega_{1,i}, \cdots, \omega_{C,i}\}'$ to encode the cluster membership for account $i$, where $\omega_{c,i} = 1$ if account $i$ belongs to the $c$-th cluster and 0 otherwise. In analogy to other model-based clustering approaches, the unobserved cluster membership $\boldsymbol{\omega}_i$ are treated as missing data and assumed to follow a categorical distribution with parameter $\boldsymbol{\pi} = \{\pi_1, \cdots, \pi_C\}$, where $\pi_c$ indicates the probability that an account belongs to the $c$-th cluster. Conditional on $\boldsymbol{\pi}$, we assume that $X_i^r(t)$'s in different clusters have heterogeneous behavioral patterns, characterized by their corresponding cluster-specific multivariate Gaussian processes with mean functions $\mu_{x,c}^r(t) = \mathbb{E}[X_i^r(t)|\omega_{c,i} = 1]$ and cross covariance functions $\Gamma_{x,c}^{r,r'}(s, t) = \text{Cov}[X_i^r(s), X_i^{r'}(t)|\omega_{c,i} = 1]$, for $s, t \in [0, T]$, and $r, r' = 1, \cdots, R$. Here, $\mu_{x,c}^r(t)$ characterizes the cluster-specific first-order intensity function, and $\Gamma_{x,c}^{r,r'}(s, t)$ describes the temporal dependence patterns within and across event types in the same cluster $c$, $c = 1, \cdots, C$.

Similarly, we assume that $\mathbf{Y}_j(t) = \{Y_j^r(t)\}_{r=1}^R$ and $\mathbf{Z}_{i,j}(t) = \{Z_{i,j}(t)^r\}_{r=1}^R$ are both mean-zero multivariate Gaussian processes to account for dependence of day-level and residual random effects within and across event types, respectively. The covariance functions take the forms: $\Gamma_y^{r,r'}(t) = \text{Cov}[Y_j^r(t), Y_j^{r'}(t)]$, and $\Gamma_z^{r,r'}(t) = \text{Cov}[Z_{i,j}^r(t), Z_{i,j}^{r'}(t)]$. As the heterogeneity patterns are assumed to be mainly explained by the account-level effect $\mathbf{X}$, both $\Gamma_y^{r,r'}(t)$ and $\Gamma_z^{r,r'}(t)$ are assumed to be homogeneous across all clusters.

**A Single-level Special Case.** When $m = 1$, our data matrix $\mathcal{S}$ only has one column of event sequences. The multi-level model in (1) reduces to a single-level model:

$$\lambda_{i,1}^r(t|\Lambda_{i,1}^r) = \exp\{\Lambda_{i,1}^r(t)\}, \quad \Lambda_{i,1}^r = X_i^r(t), \quad t \in [0, T] \tag{2}$$

where $\mathbf{X}_i(t) = \{X_i^r(t)\}_{r=1}^R$ has the same model specification as in the multi-level case described earlier. We remark that it is still of importance to consider this special case that has also been studied in the literature [29], as even in this simpler case limited work has been done for the clustering of repeatedly observed marked point processes.

## 3.2 Likelihood Function

We denote the parameters concerning $\mathbf{X}_i(t)$ in cluster $c$ as $\Theta_{x,c}$ and the parameters concerning $Y_j(t)$ and $Z_{i,j}(t)$ as $\Theta_y$ and $\Theta_z$, respectively. Therefore, the parameters in model (1) consist of $\Omega = \{\boldsymbol{\pi}, \Theta_y, \Theta_z, \Theta_{x,c}, c = 1, \cdots, C\}$. When $m = 1$, $\Omega = \{\boldsymbol{\pi}, \Theta_{x,c}, c = 1, \cdots, C\}$ representing the parameters involved in model (2). The complete data $\mathcal{D}$ consists of the observed data $\mathcal{S}$ and the unobserved latent variables $\{\{\boldsymbol{\omega}_i\}_{i=1}^n, \mathcal{L}\}$, where $\mathcal{L} = \{\{\mathbf{X}_i(t)\}, \{\mathbf{Y}_i(t)\}, \{\mathbf{Z}_{i,j}(t)\}\}$ for model (1)

and $\mathcal{L} = \{\{\mathbf{X}_i(t)\}\}$ for model (2). Let $S_i$ be the $i$-th row of $\mathcal{S}$ representing activities of the $i$-th account. In our mixture model, the probability of the observed data $\mathcal{S}$ can be written as

$$p(\mathcal{S}; \Omega) = \mathbb{E}_{\omega}\mathbb{E}_{\mathcal{L}}\left[\prod_{i=1}^{n} \text{PP}(S_i|\mathcal{L}) \mid \{\boldsymbol{\omega}_i\}_{i=1}^{n}; \Omega\right], \tag{3}$$

where the expectations are taken with respect to the conditional distribution of latent variables $\mathcal{L}$ and $\boldsymbol{\omega}_i$'s, and $\text{PP}(S_i|\mathcal{L})$ is the conditional probability of a Poisson point process,

$$\text{PP}(S_i \mid \mathcal{L}) = \prod_{j=1}^{m}\prod_{r=1}^{R}\left\{\prod_{t \in S_{i,j}^r} \lambda_{i,j}^r(t \mid \Lambda_{i,j}^r) \exp\left[-\int_0^T \lambda_{i,j}^r(s \mid \Lambda_{i,j}^r)ds\right]\right\}, \tag{4}$$

where, conditional on $\mathcal{L}$, $\Lambda_{i,j}^r(t)$ has the form as (1) for $m > 1$ and as (2) for $m = 1$.

## 4   Row-clustering Algorithms

Existing mixture model-based clustering methods typically rely on likelihood-based Expectation-Maximization algorithms [1] by treating unobserved latent variables, $\{\{\boldsymbol{\omega}_i\}_{i=1}^{n}, \mathcal{L}\}$ in our case, as missing data. However, standard EM algorithms are computationally intractable for the models we consider here. One computation bottleneck is the numerical optimizations involved in M-steps, which require many iterations due to the lack of closed-form solutions when updating parameters. Moreover, the computation burden is severely aggravated by the fact that the expectations in E-step (see (3) for an example) involve an intractable multivariate integration.

In Section 4.1, we describe a novel efficient semi-parametric Expectation-Solution algorithm for the single-level model in (2) to bypass the computation challenges described above. We then show in Section 4.2 that the learning task of multi-level models in (1) can be transformed and solved by utilizing an algorithm similar to that of single-level models.

### 4.1   Learning of Single-level Models

The ES algorithm [6] is a general iterative approach to solving estimating equations involving missing data or latent variables. The algorithm proceeds by first constructing estimating equations based on a complete-data summary statistic, which may arise from a likelihood, a quasi-likelihood, or other generalized estimating equations. Similar to the EM algorithm, the ES algorithm then iterates between an expectation (E)-step and a solution (S)-step until convergence to obtain parameter estimates. The detailed steps of a general ES algorithm framework are included in Supplementary S.2. The EM framework is a special case of ES when estimating equations are constructed from full likelihoods and using complete data as the summary statistic.

Due to the lack of closed-form for the likelihood function (3), we opt to design our algorithm under the more flexible and general ES framework for parameter estimations of the single-level models in (2), i.e., $m = 1$. The algorithm is summarized in Algorithm 1 and detailed below.

As a preliminary, we give the form of the expectation of the conditional intensity function given cluster memberships as follows:

$$\rho_c^r(t) = \mathbb{E}[\lambda_{i,1}^r(t) \mid \omega_{c,i} = 1] = \exp[\mu_{x,c}^r(t) + \Gamma_{x,c}^r(t,t)/2]. \tag{5}$$

The form of the second-order conditional intensity function is

$$\rho_{c,i}^{r,r'} = \mathbb{E}[\lambda_i^r(s)\lambda_i^{r'}(t) \mid \omega_{c,i} = 1] = \mathbb{E}\{\exp[X_i^r(s) + X_i^{r'}(t)|\omega_{c,i} = 1]\}$$
$$= \rho_c^r(s)\rho_c^{r'}(t)\exp[\Gamma_{x,c}^{r,r'}(s,t)], \tag{6}$$

for $i = 1, \cdots, n, r, r' = 1, \cdots, R$, where the last equality is derived following the moment generating function of a Gaussian random variable.

**Estimating Equations.** We carefully construct estimating equations of unknown parameters with three considerations in mind: (1) the expectation of the estimating equations over the complete data should be zero; (2) the conditional expectation of the estimating equation can be solved efficiently in the S-step; (3) the estimating equations should be fast to calculate.

Let $K(\cdot)$ be a kernel function and $K_h(t) = h^{-1}K(t/h)$ with a bandwidth $h > 0$. We define

$$A_c^{r,r'}(s,t;h) = \sum_{i=1}^{n} \omega_{c,i} a_i^{r,r'}(s,t;h), \quad \text{where } a_i^{r,r'}(s,t;h) = \sum_{u \in S_i^r, v \in S_i^{r'}}^{u \neq v} \frac{K_h(s-u)K_h(t-v)}{ng(s;h)g(t;h)};$$

$$B_c^r(t;h) = \sum_{i=1}^{n} \omega_{c,i} b_i^r(t;h), \qquad \text{where } b_i^r(t;h) = \sum_{u \in S_i^r} \frac{K_h(t-u)}{ng(t;h)}, \tag{7}$$

for $c = 1, ..., C$, and $r, r' = 1, ..., R$, where $g(x;h) = \int K_h(x-t)dt$. Using the Campbell's Theorem [17] and the moment generating function of the normal distribution, it is straightforward to show that $\mathbb{E}\left[A_c^{r,r'}(s,t;h)|\boldsymbol{\omega}\right] \approx \pi_c \rho_c^r(s) \rho_c^{r'}(t) \exp[\Gamma_{x,c}^{r,r'}(s,t)]$ and that $\mathbb{E}[B_c^r(t;h)|\boldsymbol{\omega}] \approx \pi_c \rho_c^r(t)$, provided that $h$ is sufficiently small. This motivates us to consider the following estimating equations:

$$\begin{cases} A_c^{r,r'}(s,t;h) - \pi_c \rho_c^r(s)\rho_c^{r'}(t)\exp[\Gamma_{x,c}^{r,r'}(s,t)] = 0 \\ B_c^r(t;h) - \pi_c \rho_c^r(t) = 0 \\ n^{-1}\sum_{i=1}^{n} \omega_{c,i} - \pi_c = 0. \end{cases} \tag{8}$$

**Expectation (E-step).** Given an observed data $\mathcal{S}$ and a current parameter estimate $\Omega^*$, we calculate the conditional expectation of the estimation equations in (8). Note that, conditioned on $\mathcal{S}$, the three estimating equations are all linear with respect to $\{\omega_{c,i}, c = 1, \cdots, C, i = 1, \cdots, n\}$. Therefore, the conditional expectations of the estimating equations are obtained by replacing $w_{c,i}$ with its conditional expectation $\mathbb{E}_{\boldsymbol{\omega}}[\omega_{c,i}|\mathcal{S}; \Omega^*]$, which has the following form:

$$\mathbb{E}_{\boldsymbol{\omega}}[\omega_{c,i}|\mathcal{S}; \Omega^*] = \frac{\pi_c^* f(S_i|\omega_{c,i}=1; \Omega^*)}{\sum_{c=1}^{C} \pi_c^* f(S_i|\omega_{c,i}=1; \Omega^*)}, \tag{9}$$

where $f(S_i|\omega_{c,i}=1; \Omega^*) = \mathbb{E}_{\mathcal{L}}[\text{PP}(S_i|\mathcal{L})|\omega_{c,i}=1; \Omega^*]$. Here $\text{PP}(\cdot)$ is the conditional distribution function of $S_i$ given $\omega_{c,i}$ and $\Omega^*$ as defined in (4). We propose to approximate $f(S_i|\omega_{c,i}=1; \Omega^*)$ by its Monte Carlo counterpart,

$$\hat{f}(S_i \mid \omega_{c,i}=1; \Omega^*) \approx Q^{-1}\sum \hat{\text{PP}}(S_i \mid \boldsymbol{X}_c^{(q)}(t)), \tag{10}$$

where $Q$ is the Monte Carlo sample size, $\boldsymbol{X}_c^{(q)}(t)$'s are independent samples from the multivariate Gaussian process with parameters $\Theta_{x,c}^*$ (see details below), and $\hat{\text{PP}}(\cdot)$ is a numerical quadrature approximation of $\text{PP}(\cdot)$ following [2]:

$$\hat{\text{PP}}(S_i|\boldsymbol{X}(t)) = \exp\left\{\sum_{r=1}^{R} \sum_{u \in \tilde{S}_{i,1}^r} v_u[y_u X^r(u) - \exp X^r(u)]\right\}, \tag{11}$$

In the above, $\tilde{S}_{i,1}^r$ is the union of $S_{i,1}^r$ and a set of regular grid points, $v_u$ is the quadrature weight corresponding to each $u$ and $y_u = v_u^{-1}\Delta_u$, where $\Delta_u$ is an indicator of whether $u$ is an observation ($\Delta_u = 1$) or a grid point ($\Delta_u = 0$).

**Solution (S-step).** In this step, we update the parameters by finding the solutions to the expected estimating equations from the E-step. For $c = 1, \cdots, C, r, r' = 1, \cdots, R$ and $r \neq r'$, the solutions take the following closed forms:

$$\pi_c^* = n^{-1}\sum_{i=1}^{n} \mathbb{E}[\omega_{c,i}|\mathcal{S}; \Omega^*], \tag{12}$$

$$\Gamma_{x,c}^{r,r'*}(s,t) = \log \frac{\pi_c^* \mathbb{E}_{\boldsymbol{\omega}}[A_c^{r,r'}(s,t;h)|\mathcal{S}; \Omega^*]}{\mathbb{E}_{\boldsymbol{\omega}}[B_c^r(s;h)|\mathcal{S}; \Omega^*]\mathbb{E}_{\boldsymbol{\omega}}[B_c^{r'}(t;h)|\mathcal{S}; \Omega^*]}, \tag{13}$$

$$\mu_{x,c}^{r*}(t) = \log\left\{\pi_c^{*-1}\mathbb{E}_{\boldsymbol{\omega}}[B_c^r(t;h)|\mathcal{S}; \Omega^*]/\exp[\Gamma_{x,c}^{r*}(t,t)/2]\right\}. \tag{14}$$

**Sampling Strategies.** The multi-dimensional functional form of $\boldsymbol{X}_c^{(g)}$ renders the sampling procedures in (10) intractable. Given $\Theta_{x,c}$, one solution is to find a low-rank representation of $\boldsymbol{X}_i$ with the functional principal components analysis (FPCA) [20]. Specifically, we approximate the latent Gaussian process $\boldsymbol{X}_i$ in cluster $c$ nonparametrically using the Karhunen-Lòeve expansion [25] as: $X_i^r(t) = \boldsymbol{\mu}_c + \boldsymbol{\xi}_i^T \boldsymbol{\phi}(t)$, for $r = 1, \cdots, R$, where $\boldsymbol{\xi}_i$ is a vector of normal random variables, and $\boldsymbol{\phi}(t)$ is a vector of orthogonal eigenfunctions. Using FPCA, we can obtain the samples of $\boldsymbol{X}_i$ by sampling $\boldsymbol{\xi}_i$ indirectly. More detailed sampling procedure via FPCA can be seen in Supplementary S.2.

**Remarks.** The most significant advantage of our method is that it avoids expensive iterations inside each E-step and S-step, unlike other EM algorithms for mixture point process models [30]. The elements $a_i^{r,r'}(s,t;h)$'s and $b_i^r(t;h)$'s in (7) can be pre-calculated before E-S iterations to save computations. Moreover, the S-step is fast to execute thanks to the closed-form solutions. We will analyze the overall computation complexity of the learning algorithm in Section 4.3.

---

**Algorithm 1** Learning of the Single-level model in (2)

---

**Input:** $\mathcal{S} = \{S_i\}_{i=1}^n$, the number of clusters $C$, the bandwidth $h$;
**Output:** Estimates of model parameters, $\hat{\boldsymbol{\pi}}, \hat{\mu}_{x,c}^r(t), \hat{\Gamma}_{x,c}^{r,r'}(s,t)$, for $c = 1, \cdots, C, r, r' = 1, \cdots, R$;
Calculate the components $a_i^{r,r'}(s,t;h)$'s and $b_i^r(t;h)$'s given in (7);
Initialize $\Omega^* = \{\boldsymbol{\pi}^*, \Theta_{x,c}^*, c = 1, \cdots, C\}$ randomly;
**Repeat:**
    *E-Step:*
    Calculate $\mathbb{E}_{\boldsymbol{\omega}}[\omega_{c,i}|\mathcal{S};\Omega^*]$ as (9);
    Calculate $\mathbb{E}_{\boldsymbol{\omega}}[A_c^{r,r'}(s,t)|\mathcal{S};\Omega^*]$ and $\mathbb{E}_{\boldsymbol{\omega}}[B_c^r(t)|\mathcal{S};\Omega^*]$ as linear combinations of $\mathbb{E}_{\boldsymbol{\omega}}[\omega_{c,i}|\mathcal{S};\Omega^*]$'s.
    *S-Step:*
    Update $\boldsymbol{\pi}^*, \mu_{x,c}^{r*}(t)$ and $\Gamma_{x,c}^{r,r'*}(s,t)$ according to (12), (13) and (14);
    **End;**
**Until:** Reach the convergence criteria;
$\hat{\boldsymbol{\pi}} = \boldsymbol{\pi}^*, \hat{\mu}_{x,c}^r = \mu_{x,c}^{r*}(t)$ and $\hat{\Gamma}_{x,c}^{r,r'}(s,t) = \Gamma_{x,c}^{r,r'*}(s,t)$;

---

**Model Selection.** Algorithm 1 requires choosing a proper number of clusters $C$ and bandwidth $h$. In model-based clustering, one popular method for choosing the number of clusters is based on the Bayes information criterion (BIC) [22], which can be readily computed for our model since the probability $f(\mathcal{S}|\boldsymbol{\omega})$ is already calculated in each iteration. The choice of bandwidth $h$ also plays an important role in model estimation. A small $h$ may produce unstable clustering results while a large $h$ would dampen the characteristics of each cluster. With $a_i^{r,r'}(s,t;h)$'s and $b_i^r(s;h)$'s given in (7) pre-calculated for different candidates of $h$, we can adaptively choose the $h$ that maximizes the likelihood in each iteration in a computationally efficient manner.

### 4.2 Learning of Multi-level Models

We now consider developing the learning algorithm for the multi-level model (1), assuming we repeatedly observe $R$ types of events from $n$ accounts on $m$ days with $m > 1$. Below, we propose a method to transform the learning task of a multi-level model into a problem that can be solved by a two-step procedure, where the second step is mathematically equivalent to a single-level model and hence can be conveniently solved by a similar algorithm as in Algorithm 1.

For a given account $i$, consider the aggregated event sequence $\bar{S}_{i.}^r = \cup_{j=1}^m S_{i,j}^r$ for each row of $\mathcal{S}$ and event type $r$. If we assume a multi-level model for each $S_{i,j}^r$ as in (1), conditional on latent variables $\mathcal{L}$, $\bar{S}_{i.}^r$ is a superposition of $m$ independent Poisson processes and hence can be viewed as a new Poisson process with intensity functional $\lambda_{i.}^r(t|\mathcal{L}) = \sum_{j=1}^m \exp \Lambda_{i,j}^r(t)$. We approximate the distribution of $\bar{S}_{i.}^r$ by a Poisson process with a marginal intensity function,

$$\bar{\lambda}_i^r(t) = \mathbb{E}_{YZ}\{\lambda_{i.}^r(t|\mathcal{L})|X_i(t)\} = m \exp\{\tilde{X}_i^r(t)\}, \tag{15}$$

where $\tilde{\boldsymbol{X}}_i = \{\tilde{X}_i^1, \cdots, \tilde{X}_i^R\}$ is a new multivariate mixture Gaussian process with mean function $\tilde{\mu}_{x,c}^r(t) = \mu_{x,c}^r + \Gamma_y^{r,r}(t,t)/2 + \Gamma_z^{r,r}(t,t)/2$ and covariance function $\tilde{\Gamma}_{x,c}^{r,r'}(s,t) = \Gamma_{x,c}^{r,r'}(s,t)$, if account $i$ belongs to cluster $c$. When $m$ is large, we expect the above approximation to be accurate.

Note that the model in (15) for the aggregated event sequence $\bar{S}_{i.}^r$ is inherently reduced to a single-level model. It allows us to separate the inference of the multi-level model in (1) into two steps: (*Step I*) learn the parameters in $\Theta_y$ and $\Theta_z$ and denote the estimated parameters as $\hat{\Gamma}_y^{r,r'}(s,t)$ and $\hat{\Gamma}_z^{r,r'}(s,t)$; (*Step II*) learn the clusters of the single-level model in (15) and estimate the parameters $\boldsymbol{\pi}$, $\tilde{\mu}_{x,c}^r$ and $\tilde{\Gamma}_{x,c}^{r,r'}$. Afterwards, the parameters involved in $\Theta_{x,c}$ can be obtained by

$$\hat{\mu}_{x,c}^r(t) = \tilde{\mu}_{x,c}^r(t) - \hat{\Gamma}_y^{r,r}(t,t)/2 - \hat{\Gamma}_z^{r,r}(t,t)/2, \quad \hat{\Gamma}_{x,c}^{r,r'}(s,t) = \tilde{\Gamma}_{x,c}^{r,r'}(s,t).$$

For the learning task in Step I, [28] developed a semi-parametric algorithm to learn the repeatedly observed event sequences. In analogy to their work, we propose a similar inference framework to estimate $\Theta_y$ and $\Theta_x$ in our mixture multi-level model (1) and provide the details in Supplementary S.1. For step II, we resort to the single-level model algorithm described in Section 4.1.

### 4.3 Computational Complexity and Acceleration

Assume that the training event sequences belong to $n$ accounts and $C$ clusters and are repeatedly observed on $m$ time slots. We also assume that the data contains $R$ types of events and each sequence consists of $I$ time stamps on average. Let $Q$ be the sampling size used in the Monte Carlo integration in (10). In numerical implementation, we divide the interval $[0, T]$ into $D$ equally spaced grid points $\mathcal{D} = \{0 = u_1 < \cdots < u_D = T\}$. In Step I, it requires $O(nmR^2D^2)$ computation complexity to estimate $\Theta_y$ and $\Theta_z$, according to [28]. Computation complexity to pre-calculate $a_i^{r,r'}(s,t;h)$'s and $b_i^r(t;h)$'s in (7) for all $s,t \in \mathcal{D}$ is of the order $O(nmR^2D^2)$ if we decomposition $a_i^{r,r'}(s,t;h)$ as:

$$a_i^{r,r'}(s,t;h) = \left[\sum_{u \in S_i^r} \frac{K_h(s-u)}{g(s;h)}\right]\left[\sum_{v \in S_i^{r'}} \frac{K_h(t-v)}{g(t;h)}\right] - \sum_{u \in S_i^r \cap S_i^{r'}} \frac{K_h(s-u)K_h(t-v)}{g(s;h)g(t;h)}.$$

In Step II, for each E-S iteration, we need $O(CQR^3)$ for sampling and $O(nCIQR^2)$ for other calculations. Therefore, the overall computational complexity is $O(R^2(nmD^2 + CQR + nCIQ))$. To further reduce computation, we use array programming and GPU acceleration to calculate the high-dimensional integration in the Monte Carlo EM framework [26] to reduce the runtime of (9). The details are included in Supplementary S.2, and a numerical demonstration is given in Section 5.1.

## 5 Numerical Examples

We examine the performance of our **MM-MPP** framework for clustering event sequences via synthetic data examples and real-world applications and compare the performances between the proposed method and two other state-of-the-art methods. One competing method is a discrete Fréchet distance-based method (**DF**) by [18]. Unlike other distance-based clustering methods, the DF cluster can characterize interactions among events. Another is a model-based clustering method based on the Dirichlet mixture of Hawkes processes (**DMHP**) by [30]. DMHP is chosen as a competitor due to its capability of accounting for complex point patterns while performing clustering and making efficient variational Bayesian inference algorithms under a nested EM framework.

### 5.1 Synthetic Data

**Setting.** We generate the synthetic data from the proposed mixture model of log-Gaussian Cox processes in (1) and (2), in which there are $R = 5$ event types and daily time stamps reside in $[0, 2]$. We set the number of clusters $C$ from 2 to 5 and set the number of accounts in each cluster to 500. We experiment with an increasing number of replicates ($m = 1$, 20 or 100), to check the convergence of our method. When $m = 1$, we generate event sequences from the single-level model in (2) without day-level variations. In this case, we compare the clustering results of DF, DMHP with those of the single-level model (MS-MPP). When $m = 20$ or 100, we generate data from the multi-level model in (1) and use the MM-MPP method to model the scenario where event sequences are repeatedly observed. However, the two competing methods, DF and DMHP, are not directly applicable for repeated event sequences. Therefore, in this case, we concatenate $\{S_{i,j}^r\}_{j=1}^m$ sequentially into a new event sequence $S_{i.}^r$ on $[0, mT]$ and then apply DF and DMGP to this new sequence. The detailed

settings of $X_i^r(t)$'s, $Y_j^r(t)$'s and $Z_{i,j}^r(t)$'s and other details of synthetic data examples are elaborated in Supplementary S.3.

**Results.** We evaluate the clustering performance of each method over 100 repeated experiments under each setting, using *clustering purity* [21] as a evaluation metric. Table 1 reports the averaged clustering purity of each method on the synthetic data. When $m = 1$, MS-MPP obtains the best clustering result in terms of purity consistently across different numbers of clusters. Especially when $C$ increases, in which case there are more overlaps among clusters, the advantage of MS-MPP becomes more prominent. When $m = 20, 100$, MM-MPP still significantly outperforms the other two competitors. It is also noticeable that the performance of DF and DMPH, in general, deteriorates as $m$ increases, although more repeated event sequences offer more information for clustering. One explanation is that both DF and DMPH may incur bias due to ignoring different sources of variations for repeatedly observed event times. Another reason may be that many existing Hawkes process models, such as DMPH, assume a constant triggering function over time, which may not be flexible enough to characterize the data generated from models (1) and (2).

Table 1: Clustering Purity on Synthetic Data.

| | $m = 1$ | | | $m = 20$ | | | $m = 100$ | | |
|---|---|---|---|---|---|---|---|---|---|
| $C$ | DF | DMPH | MS-MPP | DF | DMPH | MM-MPP | DF | DMPH | MM-MPP |
| 2 | 0.597 | 0.537 | **0.831** | 0.536 | 0.513 | **0.947** | 0.532 | 0.522 | **0.988** |
| 3 | 0.514 | 0.466 | **0.767** | 0.465 | 0.423 | **0.902** | 0.477 | 0.394 | **0.967** |
| 4 | 0.443 | 0.421 | **0.714** | 0.422 | 0.356 | **0.874** | 0.436 | 0.285 | **0.944** |
| 5 | 0.379 | 0.354 | **0.675** | 0.351 | 0.298 | **0.835** | 0.333 | 0.276 | **0.919** |

Our code can be accessed via `https://github.com/LihaoYin/MMMPP`. To show the computational advantage of the proposed ES algorithm over the EM algorithm, Table 2 gives the computation times of CPU-based EM, CPU-based ES, and GPU-based ES algorithms for 20 iterations in the estimation of model (1) with $n = 500, 100$, $m = 20$, $R = 5$ and $C = 3$. For each iteration, $10,000$ MCMC samples are drawn to approximate (10). Table 2 demonstrates that with the GPU acceleration, the computation time of the proposed ES can be reduced by more than 20 folds in this case scenario compared to the EM algorithm, which is not suitable for array programming [7].

Table 2: Running Time (in seconds) on Synthetic Data

| Methods and devices | $n = 500$ | $n = 1000$ |
|---|---|---|
| **GPU-ES** (RTX 8000 48G GPU) | **30.09** | **51.42** |
| CPU-**ES** (i7-7700HQ CPU) | 275.87 | 505.07 |
| CPU-EM (i7-7700HQ CPU) | 568.36 | 1105.46 |

## 5.2 Real-world Data

In this section, we apply our method to the following real-world datasets.

**Twitter Dataset.** The Twitter dataset consists of the postings of the official accounts of America's top 500 universities from April 15, 2021, to May 14, 2021. The data set was scraped from Twitter with the API `rtweet` [12]. The dataset involves three categories of postings (tweet, retweet, and reply), indicating $R = 3$ in this study. As a result, the dataset contains $n = 500$ Twitter accounts for $m = 30$ consecutive days with a total of $233,465$ time stamps.

**Chicago City Taxi Dataset** The City of Chicago collected the information of all taxi rides in Chicago since 2013 [1]. Each trip record in the dataset consists of drivers' encrypted IDs, pick-up/drop-off time stamps, and locations (in the form of latitude/longitude coordinates). We gathered the trips of 9,000 randomly selected taxi drivers from Jan 1 to Dec 31, 2016, and more than 19 million trip records were picked. We mapped the pick-up coordinates to their corresponding zoning types according to Chicago Zoning Map Dataset[2], which divides the city into nine basic zoning districts[3], including Residence (R), Business (B), Commercial (C), Manufacturing (M), etc. For this data set, we have $n = 9000$, $m = 366$, and $R = 9$.

---

[1] `https://data.cityofchicago.org/Transportation/Taxi-Trips/wrvz-psew`
[2] `https://data.cityofchicago.org/`
[3] `https://secondcityzoning.org/zones/`

**Credit Card Transaction Dataset.** The dataset contains $641,914$ transaction records of $5,000$ European credit card customers ($n = 5000$) during the period covering January 1 to December 31, 2016 ($m = 366$). We applied the univariate model ($R = 1$) without event marks to the dataset.

We evaluate and compare clustering stability based on a measure called *clustering consistency* via $K$-trial cross-validations [23, 24], as there are no ground truth clustering labels. The detailed definition of *clustering consistency* and other real data example details are included in Supplementary S.4.

**Results.** We compare the performance of DF, DMHP, and MM-MPP in terms of clustering consistency for three data sets with $K = 100$ trials. The results in Table 3 suggest that MM-MPP outperforms its competitors notably, demonstrating that our model can better characterize the postings patterns and offer a more stable and consistent clustering than other methods. Figure 2 shows the histograms of the number of learned clusters for each method. For the Twitter dataset, the median numbers of learned clusters are $3$, $5$, and $8$ for MM-MPP, DMHP, and DF respectively. Besides, the distribution of the number of clusters from our method seems to be the least variable, indicating robustness in clustering. The robustness of our method may be partly attributed to the flexibility of the latent conditional intensity functions that account for multi-level deviations within each account. In contrast, other methods that fail to account for different sources of deviations may treat them as sources of heterogeneity and consequently result in more clusters.

Table 3: Clustering Consistency on Real-World Datasets.

| Method | DF | DMHP | MM-MPP |
|---|---|---|---|
| Twitter | 0.096 | 0.275 | **0.394** |
| Credit Card | 0.102 | 0.331 | **0.378** |
| Chicago Taxi | 0.045 | 0.142 | **0.153** |

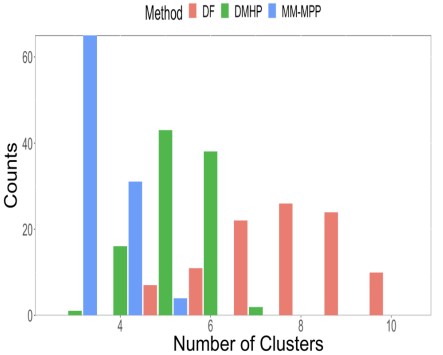 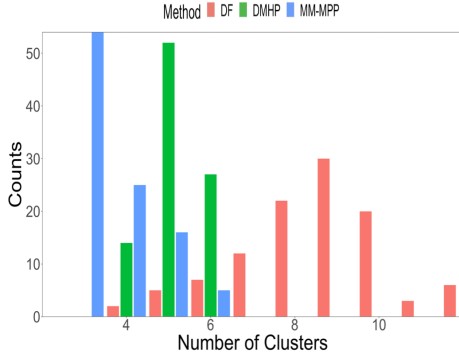

Figure 2: Histogram of the number of clusters. Left: Twitter dataset; Right: Credit Card dataset;

More stories can be told by the estimated posting patterns. Given a predicted membership of account $i$ by $c_i = \arg\max_c \mathbb{E}_\omega[\omega_{c,i}|\mathcal{S}; \hat{\Omega}]$, Figure 3 displays the estimated curves of $\hat{\mu}_{x,c}^r$ for tweet events ($r = 1$), retweet events ($r = 2$) and reply events ($r = 3$) respectively for $C = 3$. Recall $\hat{\mu}_{x,c}^r$ is interpreted as the baseline of intensity functions. This figure shows three different activity modes for the selected Twitter accounts. The universities in cluster 1 marked by red curves in Figure 3 in general have a lower frequency of posting retweets and replies, especially during the daytime. This group includes the most top university in America, such as MIT, Harvard, and Stanford. In contrast, the accounts in cluster 2 are relatively more active in all three types of postings. We further find that many accounts in this cluster belong to the universities with middle ranks.

We further applied the proposed MM-MPP to the Chicago Taxi dataset. As suggested by BIC, the 9000 taxi drivers are clustered into 9 groups, whose averaged daily pick-up log intensity functions are illustrated in Figure 4(a). We can see that the taxi drivers are clustered not only according to their pick-up frequency but also by their working schedules. For example, the black curves on Figure 4(a) corresponds to the most dominating group, which occupies 23.2% of the sample. Figure 3(b) displays the curves of average log intensity (the black line) and log intensity for each driver (gray lines) in the selected cluster. Figure 4(c-e) show the estimated $\hat{\mu}_x(t)$ for pick-up in commercial, residence and

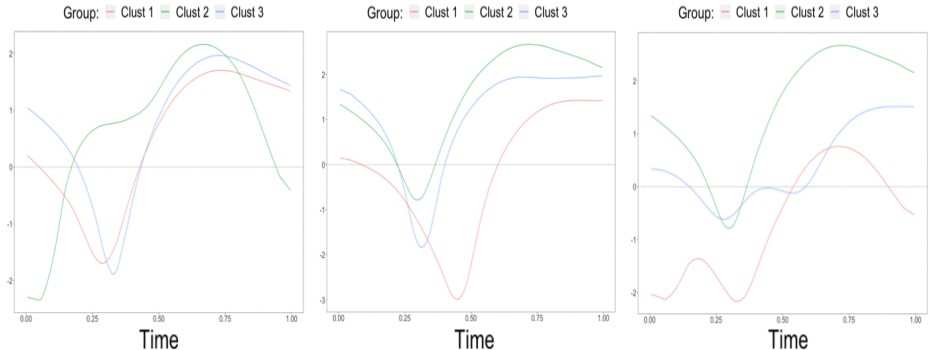

Figure 3: Curves of $\hat{\mu}_{x,c}^r(t)$. Left: tweet events; Mid: retweet events; Right: reply events;

manufacturing districts, respectively. While the pick-up events are more likely to occur in commercial districts for this group during the daytime, they also tend to pick up passengers at the residential district in the morning and to appear at the manufacturing district in the afternoon. These patterns are consistent with the schedules of passengers who commute between homes and workplaces.

More results and discussions on chase credit dataset are included in our Supplementary file.

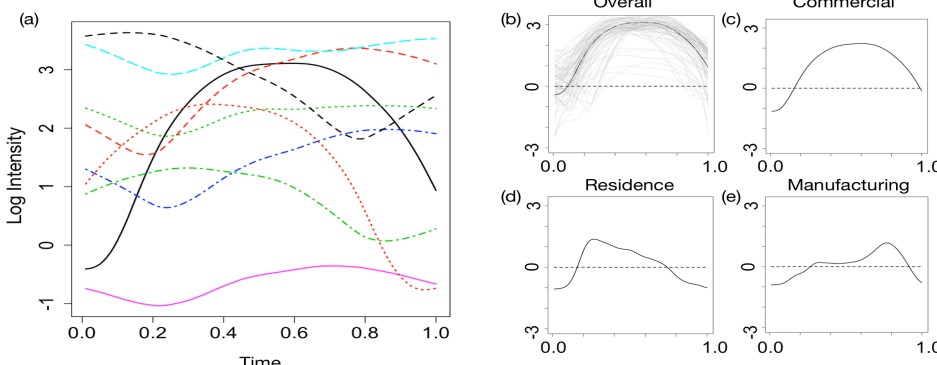

Figure 4: Left: Overall log-intensities for all clusters; Right: Log-intensity for one selected cluster;

# 6   Concluding Remarks

We propose a mixture of multi-level marked point processes to cluster repeatedly observed marked event sequences. A novel and efficient learning algorithm is developed based on a semi-parametric ES algorithm. The proposed method is demonstrated to significantly outperform other competing methods in simulation experiments and real data analyses.

The current model only focuses on events over temporal domains. However, clustering of spatial patterns on 2- or 3-dimensional domains has also attracted much research interest [10, 31, 9]. It will be an interesting research topic to extend the current model to such settings.

This work has no foreseeable negative societal impacts, but users should be cautious when giving interpretation on clustering results to avoid any misleading conclusions.

## Acknowledgement

We thank the anonymous reviewers for their constructive comments that help improve the manuscript significantly. Xu's research is supported by NSF grant SES-1902195, Guan's research is supported by NSF grant SES-1758575, and Sang's research is supported by NSF grant DMS-1854155.

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
