# Supplementary of "Row-clustering of a Point Process-valued Matrix"

**Lihao Yin**[1,2]     **Ganggang Xu**[3]     **Huiyan Sang**[2]     **Yongtao Guan**[3]

[1]Institute of Statistics and Big Data, Renmin University, Beijing, China,

[2]Department of Statistics, Texas A&M University, College Station, Texas,

[3]University of Miami, Coral Gables, Florida,

{lihao, huiyan}@stat.tamu.edu, {gangxu, yguan}@bus.miami.edu

## Abbreviations

- **ES**: Expectation-Solution;
- **LGCP**: log-Gaussian Cox process;
- **FPCA**: Functional principal component analysis;
- **MM-MPP**: Mixture Multi-level Marked Point Processes;
- **MS-MPP**: Mixture Single-level Marked Point Processes;
- **MC**: Monte Carlo
- **DMHP**: Dirichlet mixture of Hawkes processes;
- **DF**: discrete Fréchet;

## S.1    Step I of the Two-step Learning of the Multi-Level Model

We consider a multi-level model with the following latent intensity function:
$$\lambda_{i,j}^r(t) = \exp\{X_i^r(t) + Y_j^r(t) + Z_{i,j}^r(t)\}, \quad t \in [0, T] \tag{S1}$$
for $i = 1, \cdots, n$, $j = 1, \cdots, m$ and $r = 1, \cdots, R$.

As discussed in Section 4.2, the learning algorithm is decomposed into two steps as in Algorithm S.1. In Step I, we seek to estimate the parameters in $\Theta_y$ and $\Theta_z$. Other cluster-specific model parameters such as cluster assignment probabilities $\boldsymbol{\pi}$ are estimated in Step II following the procedure described in Section 4.2. [10] developed a semi-parametric algorithm to estimate the covariance functions of a multi-level log-Gaussian Cox process. We extend their estimation method to also take into account unknown clustering when estimating $\Theta_y$ and $\Theta_z$ in Step I. Interestingly, we will show that the resulting estimators of $\Theta_y$ and $\Theta_z$ do not depend on any other cluster-specific parameters and hence avoid iterations between the two steps.

Specifically, following the formula of the moment generating function of a Gaussian random variable, the marginal intensity functions can be calculated as
$$\rho^r(t) = \mathbb{E}[\lambda_{i,j}^r(t)] = \sum_{c=1}^C \pi_c \exp\{\mu_{x,c}^r(t) + \Gamma_{x,c}^{r,r}(t,t)/2 + \Gamma_y^{r,r}(t,t)/2 + \Gamma_z^{r,r}(t,t)/2\},$$
and derived in a similar way, the marginal second-order intensity functions are:
$$\begin{aligned}
\rho_{i,j,i',j'}^{r,r'}(s,t) &= \mathbb{E}[\lambda_{i,j}^r(s)\lambda_{i',j'}^{r'}(t)] \\
&= \sum_c \sum_{c'} \mathbb{E}[\exp\{Y_j^r(s) + Y_{j'}^{r'}(t) + Z_{i,j}^r(s) + Z_{i',j'}^{r'}(t)\}] \\
&\quad \cdot \mathbb{E}[\omega_{c,i}\omega_{c',i'}] \cdot \mathbb{E}[\exp\{X_i^r(s) + X_{i'}^{r'}(t)\}|\omega_{c,i} = 1, \omega_{c',i'} = 1]
\end{aligned}$$

35th Conference on Neural Information Processing Systems (NeurIPS 2021).

for $i, i' = 1, \cdots, n$, $j, j' = 1, \cdots, m$ and $r, r' = 1, \cdots, R$.

We analyze the form of $\rho_{i,j,i',j'}^{r,r'}$ under four different situations and use $A^{r,r'}$, $B^{r,r'}$, $C^{r,r'}$ or $D^{r,r'}$ to represent its form under each situation respectively,

$$\rho_{i,j,i',j'}^{r,r'}(s,t) =$$
$$\begin{cases} A^{r,r'}(s,t) \equiv \exp\{\Gamma_y^{r,r'}(s,t) + \Gamma_z^{r,r'}(s,t)\} \sum_c \pi_c \rho_c^r(s) \rho_c^{r'}(t) \exp\{\Gamma_{x,c}^{r,r'}(s,t)\}, & \text{if } i = i', j = j' \\[2mm] B^{r,r'}(s,t) \equiv \sum_c \pi_c \rho_c^r(s) \rho_c^{r'}(t) \exp\{\Gamma_{x,c}^{r,r'}(s,t)\}, & \text{if } i = i', j \neq j' \\[2mm] C^{r,r'}(s,t) \equiv \exp\{\Gamma_y^{r,r'}(s,t)\} \sum_{c,c'} \pi_c \pi_{c'} \rho_c^r(s) \rho_{c'}^{r'}(t), & \text{if } i \neq i', j = j' \\[2mm] D^{r,r'}(s,t) \equiv \sum_{c,c'} \pi_c \pi_{c'} \rho_c^r(s) \rho_{c'}^{r'}(t), & \text{if } i \neq i', j \neq j' \end{cases}$$
$$\text{(S2)}$$

It can be seen that $A^{r,r'}(s,t)$, $B^{r,r'}(s,t)$, $C^{r,r'}(s,t)$ and $D^{r,r'}(s,t)$ captures different correlation information, namely, the correlation within same-account same-day, within same-account across different-day, within same-day across different-account, and across different-account different-day, while integrating out the unknown cluster memberships of $i$ and $i'$.

Following a similar derivation as [10], the corresponding empirical kernel estimate of $\rho_{i,j,i',j'}^{r,r'}$ under each situation is given by

$$\begin{cases} \hat{A}^{r,r'}(s,t;h) = \sum_{i=1}^n \sum_{j=1}^m \sum_{u \in S_{i,j}^r, v \in S_{i,j}^{r'}}^{u \neq v} \frac{K_h(s-u)K_h(t-v)}{nmg(s;h)g(t;h)} \\[4mm] \hat{B}^{r,r'}(s,t;h) = \sum_{i=1}^n \sum_{j=1}^m \sum_{j' \neq j} \sum_{u \in S_{i,j}^r} \sum_{v \in S_{i,j'}^{r'}} \frac{K_h(s-u)K_h(t-v)}{nm(m-1)g(s;h)g(t;h)} \\[4mm] \hat{C}^{r,r'}(s,t;h) = \sum_{i=1}^n \sum_{i' \neq i} \sum_{j=1}^m \sum_{u \in S_{i,j}^r} \sum_{v \in S_{i',j}^{r'}} \frac{K_h(s-u)K_h(t-v)}{n(n-1)mg(s;h)g(t;h)} \\[4mm] \hat{D}^{r,r'}(s,t;h) = \sum_{i=1}^n \sum_{i' \neq i} \sum_{j=1}^m \sum_{j' \neq j} \sum_{u \in S_{i,j}^r} \sum_{v \in S_{i',j'}^{r'}} \frac{K_h(s-u)K_h(t-v)}{n(n-1)m(m-1)g(s;h)g(t;h)} \end{cases} \quad \text{(S3)}$$

for $r, r' = 1, \cdots, R$, where $K_h(t) = h^{-1}K(t/h)$ is a kernel function with bandwidth $h$ and $g(x;h) = \int K_h(x-t)dt$ is an edge correction term.

Matching (S2) with (S3), we propose to estimate the covariance functions using,

$$\hat{\Gamma}_y^{r,r'}(s,t;h) = \log \frac{\hat{C}^{r,r'}(s,t;h)}{\hat{D}^{r,r'}(s,t;h)}, \quad \hat{\Gamma}_z^{r,r'}(s,t;h) = \log \frac{\hat{A}^{r,r'}(s,t;h)\hat{D}^{r,r'}(s,t;h)}{\hat{B}^{r,r'}(s,t;h)\hat{C}^{r,r'}(s,t;h)} \quad \text{(S4)}$$

**Algorithm S.1** Learning of the Multi-level model (1)

---

**Input:** $\mathcal{S} = \{S_{i,j}^r\}$, the number of clusters $C$, the bandwidth $h$;

**Output:** Estimates of model parameters, $\hat{\boldsymbol{\pi}}, \hat{\Theta}_y, \hat{\Theta}_z, \hat{\Theta}_{x,c}$, for $c = 1, \cdots, C$;

**Step I:** Given $\mathcal{S}$, obtain $\hat{\Theta}_y$ and $\hat{\Theta}_z$ using the estimation framework in Section S.1;

**Step II:**

   a) Aggregate the event sequences by $\bar{S}_{i\cdot}^r = \cup_{j=1}^m S_{i,j}^r$;

   b) Based on $\{\bar{S}_{i\cdot}^r\}_{i=1}^n$ from a), fit the single-level model with parameters $\{\boldsymbol{\pi}, \tilde{\Theta}_{x,c}\}$ using Algorithm 1;

   c) Calculate,
   $$\hat{\mu}_{x,c}^r(t) = \tilde{\mu}_{x,c}^r(t) - \hat{\Gamma}_y^{r,r}(t,t)/2 - \hat{\Gamma}_z^{r,r}(t,t)/2 - \log m, \quad \hat{\Gamma}_{x,c}^{r,r'}(s,t) = \tilde{\Gamma}_{x,c}^{r,r'}(s,t)$$

---

## S.2   Computational Details

### S.2.1   ES Algorithm

The Expectation-Solution (ES) algorithm [2, 5] is a general extension of the Expectation-Maximization (EM) algorithm. It is an iterative approach built upon estimating equations that involve missing data or unobserved variables. In the E-step of each iteration, ES calculates the conditional expectations of estimating equations given observed data and current parameter estimates. In S-step, it updates parameter values by finding the solutions to the expected estimating equations. Since the estimating equations can be constructed from a likelihood, a quasi-likelihood, or other forms, the ES algorithm is more flexible and general than the EM algorithm. In particular, when estimating equations are well designed such that analytical solutions are available in S-step, ES algorithm may achieve an improved computational efficiency over EM algorithms, which often involve expensive numerical optimizations of the expected log-likelihood in each M-step.

We follow the notations and expressions in [2]. Let $\boldsymbol{y}$ denote the observed data vector, $\boldsymbol{z}$ denote the unobserved data, and $\boldsymbol{x} = \{\boldsymbol{y}, \boldsymbol{z}\}$ be the complete-data. Let $\boldsymbol{\Omega}$ denote a $d$-dimensional vector of parameters. Given $d$-dimensional estimating equations with the complete data as:
$$U_c(\boldsymbol{x}; \boldsymbol{\Omega}) = \mathbf{0}$$

the ES algorithm entails a linear decomposition like:

$$
\begin{aligned}
U_c(\boldsymbol{x}; \boldsymbol{\Omega}) &= U_1(\boldsymbol{y}, \boldsymbol{S}(\boldsymbol{x}); \boldsymbol{\Omega}) \\
&= \sum_{j=1}^q \boldsymbol{a}_j(\boldsymbol{\Omega}) S_j(\boldsymbol{x}) + \boldsymbol{b}_\Omega(\boldsymbol{y}),
\end{aligned}
\tag{S5}
$$

where $\boldsymbol{a}_j$'s are vectors of size $d$ only depending on parameters $\boldsymbol{\Omega}$, and $\boldsymbol{S}$ is a $q$-dimensional function with components $S_j$ only depending on the complete data. $\boldsymbol{S}(\boldsymbol{x})$ is referred to as a "complete-data summary statistic". Given the parameters $\boldsymbol{\Omega}^*$, we calculate the expectation over $\boldsymbol{z}$ condition on $\boldsymbol{y}$ and parameters $\boldsymbol{\Omega}$ in E-step as following
$$h(\boldsymbol{y}; \boldsymbol{\Omega}^*) = \mathbb{E}_z[\boldsymbol{S}(\boldsymbol{x})|y; \boldsymbol{\Omega}^*]$$

In view of the linearity in (S5), we consider the conditionally expected estimation equations,
$$\mathbb{E}_z[U_c(\boldsymbol{x}; \boldsymbol{\Omega})|\boldsymbol{y}; \boldsymbol{\Omega}^*] = U_1(\boldsymbol{y}, h(\boldsymbol{y}; \boldsymbol{\Omega}^*); \boldsymbol{\Omega}) = \mathbf{0} \tag{S6}$$

In the S-step, we update the parameters $\boldsymbol{\Omega}$ by finding the solution to (S6). We outline the ES procedure in Algorithm S.2.1.

### S.2.2   Sampling Strategy

The E-step in Section 4.1 involves the sampling of random functions $\boldsymbol{X}_c^{(q)}$ for calculating the Monte Carlo integration in (9). Given cluster-specific parameters $\Omega_{x,c}$, our goal is to draw multiple

---

**Algorithm S.2** ES Algorithm

---

**Presupposition:** Given estimating equations $U_c(\boldsymbol{x}; \boldsymbol{\Omega})$ with a linear decomposition (S5);
**Input:** Observed data $\boldsymbol{y}$;
**Output:** Estimates of model parameters $\boldsymbol{\Omega}$;
Initialize $\boldsymbol{\Omega}^*$ randomly;
**Repeat:**
    ***E-Step:***      Calculate $h(\boldsymbol{y}; \boldsymbol{\Omega}^*) = \mathbb{E}_z[\boldsymbol{S}(\boldsymbol{x})|y; \boldsymbol{\Omega}^*]$;
    ***S-Step:***      Find $\boldsymbol{\Omega}$ that solve $U_1(\boldsymbol{y}, h(\boldsymbol{y}; \boldsymbol{\Omega}^*); \boldsymbol{\Omega}) = \boldsymbol{0}$ in (S6);
    **End;**
**Until:** Reach the convergence criteria.

---

independent realizations of $\boldsymbol{X}_i(t)|\omega_{c,i} = 1$, denoted as $\boldsymbol{X}_c^{(q)}(t) = \{X_c^{1(q)}(t), \cdots, X_c^{R(q)}(t)\}'$. Recall that the cross covariance functions of $\boldsymbol{X}_i(t)$ is $\Gamma_{x,c}^{r,r'}(s,t) = \text{Cov}[X_i^r(s), X_i^{r'}(t)|\omega_{c,i} = 1]$, $s, t \in [0, T]$, for $i = 1, \cdots, n$. When $r = r'$, the covariance function $\Gamma_{x,c}^{r,r}(s,t)$ is a symmetric, continuous and nonnegative definite kernel function on $[0, T] \times [0, T]$. Then Mercer's theorem asserts that there exists the following spectral decomposition:

$$\Gamma_{x,c}^{r,r}(s,t) = \sum_{k=1}^{\infty} \eta_{x,c,k}^r \phi_{x,c,k}^r(s) \phi_{x,c,k}^r(t),$$

where $\eta_{x,c,1}^r \geq \eta_{x,c,2}^r \geq \cdots > 0$ are eigenvalues of $\Gamma_{x,c}^{r,r}(s,t)$ and $\phi_{x,c,k}^r(t)$'s are the corresponding eigenfunctions which are pairwise orthogonal in $L^2([0, T])$. The eigenvalues and eigenfunctions satisfy the integral eigenvalue equation,

$$\eta_{x,c,k}^r \phi_{x,c,k}^r(s) = \int_0^T \Gamma_{x,c}^{r,r}(s,t) \phi_{x,c,k}^r(t) dt$$

Accordingly, using the Karhunen-Loève expansion [9], $X_c^{r(q)}(t)$ admits a decomposition,

$$X_c^r(t) = \mu_{x,c}^r(t) + \sum_{k=1}^{\infty} \xi_{x,c,k}^r \phi_{x,c,k}^r(t), \tag{S7}$$

where $\{\xi_{x,c,k}^r\}_{k=1}^{\infty}$ are independent normal random variables with mean 0 and variance $\{\eta_{x,c,k}^r\}_{k=1}^{\infty}$. The expression in (S7) has an infinite dimensional parameter space, which is infeasible for estimation. One solution is to approximate (S7) by only keeping leading principal components,

$$X_c^r(t) \approx \mu_{x,c}^r(t) + \sum_{k=1}^{p_c^r} \xi_{x,c,k}^r \phi_{x,c,k}^r(t) \tag{S8}$$

where $p_c^r$ is a rank chosen to characterize the dominant characteristics of $X_c^r$ while reducing computational complexity. It leads to a reduced-rank representation of $\Gamma_{x,c}^{r,r}(s,t)$ as:

$$\Gamma_{x,c}^{r,r}(s,t) \approx \sum_{k=1}^{p_c^r} \eta_{x,c,k}^r \phi_{x,c,k}^r(s) \phi_{x,c,k}^r(t).$$

Similarly, when $r \neq r'$, we can also approximate $\Gamma_{x,c}^{r,r'}(s,t)$ using the truncated decomposition,

$$\Gamma_{x,c}^{r,r'}(s,t) \approx \sum_{k=1}^{p_c^r} \sum_{k'=1}^{p_c^{r'}} \eta_{x,c,k,k'}^{r,r'} \phi_{x,c,k}^r(s) \phi_{x,c,k}^{r'}(t). \tag{S9}$$

We denote $\boldsymbol{\xi}_{x,c}^r = \{\xi_{x,c,1}^r, \cdots, \xi_{x,c,p_c^r}^r\}'$ and investigate the cross-covariance matrix of $\boldsymbol{\xi}_{x,c} = \{\boldsymbol{\xi}_{x,c}^1, \cdots, \boldsymbol{\xi}_{x,c}^R\}$, denoted as

$$\boldsymbol{\Sigma}_{x,c} = \begin{pmatrix} \Sigma_{x,c}^{1,1} & \cdots & \Sigma_{x,c}^{1,R} \\ \vdots & \ddots & \vdots \\ \Sigma_{x,c}^{R,1} & \cdots & \Sigma_{x,c}^{R,R} \end{pmatrix},$$

where $\Sigma_{x,c}^{r,r'} = \text{Cov}[\boldsymbol{\xi}_{x,c}^r, \boldsymbol{\xi}_{x,c}^{r'}]$.

From Karhunen-Loève expansion in (S7), we know $\Sigma_{x,c}^{r,r} = \text{diag}(\eta_{x,c,1}^r, \cdots, \eta_{x,c,p_c^r}^r)$ for each event type $r$. When $c \neq c'$, we assume that $\boldsymbol{\xi}_{x,c}^r$ and $\boldsymbol{\xi}_{x,c'}^{r'}$ are independent. However, when considering two different event types (i.e., $r \neq r'$) within the same cluster, it is reasonable to account for the correlation between $\boldsymbol{\xi}_{x,c}^r$ and $\boldsymbol{\xi}_{x,c}^{r'}$ to characterize interactions among events of different types. Therefore, from (S7) and (S9), the $(k, k')$-th entry of the covariance matrix $\Sigma_{x,c}^{r,r'}$ is $\eta_{x,c,k,k'}^{r,r'}$ when $r \neq r'$.

Now we can draw the samples $\boldsymbol{\xi}_{x,c}^{(q)} = \{\boldsymbol{\xi}_{x,c}^{1(q)}, \cdots, \boldsymbol{\xi}_{x,c}^{R(q)}\}$ from the multivariate normal distribution with a mean zero and a covariance matrix is $\Sigma_{x,c}$, based on which we obtain the samples $\boldsymbol{X}_c^{(q)}$ using expansion (S8).

### S.2.3   GPU Acceleration

One computational bottleneck in our approach is the Monte Carlo (MC) approximation of the high-dimensional integration in (9). Although we have employed the low-rank representations by FPCA in Section S.2.2 to facilitate MC sampling, this step remains as the most computationally expensive part if using a naive direct calculation, due to the massive number of sampling points for a precise MC integration.

Many researchers have embarked their efforts on improving the performance of MC integration. One of the most popular frameworks is `VEGAS` [4, 6] due to its user-friendly interface. However, `VEGAS`, which is CPU-based, may be over-stretched with dimensionality going up since the required MC samples consequently increase dramatically. As notable progress, GPU-based programs, like `VegasFlow`[1], extremely boosts the computation speed compared to the CPU-version program. It accelerates the computation with the `Numpy`-like API syntax, such as `Tensorflow`, which is easy to communicate to GPU. Similar treatments are implemented in our work, and the key is to transfer the summation loop in (10) into the form of array programming [3]. For example, when we calculate $\boldsymbol{X}_c^q(t)$ in (S8), the computation involves total $p_c^r \times Q$ sampled $\xi_{x,c,k}^r$ and $p_c^r \times I \times n$ of $\phi_{x,c,k}^r(u)$, if given $c$ and $r$. It will greatly reduce the running time if we utilize array programming. For example, in the case when we have $n = 500$ sequences and $10,000$ MC points, our MS-MPP algorithm costs on average 30.09 seconds to run 20 ES iterations on RTX-8000 48G GPU. In contrast, it costs 275.87 seconds on i7-7700HQ CPU if not using array programming.

## S.3   Simulation Studies

**Setting of $X_i^r(\cdot)$'s.** In our synthetic data, we sample event sequences from $C$ heterogeneous clusters ($C = 2, 3, 4$ or $5$). Each cluster contains $500$ event sequences, and each event sequence contains $R = 5$ event types. We experiment with each setting for $J = 100$ times and investigate the average performance. In each trial, we set,

$$\mu_{x,c}^r(t) = 1 + \sum_{k=0}^{50} \zeta_k Z_{c,k}^r \cos(k\pi t) + \sum_{k=0}^{50} \zeta_k Z_{c,k}'^r \sin(k\pi t), \quad t \in [0, 2]$$

for $r = 1, \cdots, R$ and $c = 1, \cdots, C$, where $Z_{c,k}^r$'s and $Z_{c,k}'^r$'s are all independently sampled from the uniform distribution $\text{U}(-1, 1)$ and $\zeta_k = (-1)^{k+1}(k+1)^{-2}$. We set the covariance function of $X_i^r(t)$ as,

$$\Gamma_{x,c}^{r,r}(s, t) = \sum_{k=1}^{50} \tilde{Z}_{c,k}^r |\zeta_k| \sin(k\pi s + \pi \tilde{Z}_{c,k}^r) \sin(k\pi t + \pi \tilde{Z}_{c,k}^r)$$

for $r = 1, \cdots, R$ and $c = 1 \cdots, C$, where $\tilde{Z}_{c,k}^r$'s are independently sampled from uniform distribution $\text{U}(0, 0.3)$. Meanwhile, we set the interventions among different event types as,

$$\Gamma_{x,c}^{r,r'}(s, t) = \sum_{k=1}^{50} \sum_{k'=1}^{50} \check{Z}_{c,k,k'}^{r,r'} \sqrt{\tilde{Z}_{c,k}^{[j]} \tilde{Z}_{c,k'}^{[j']} |\zeta_k \zeta_{k'}|} \sin(k\pi s + \pi \tilde{Z}_{c,k}^r) \sin(k\pi t + \pi \tilde{Z}_{c,k}^r),$$

for $r \neq r'$, where $\check{Z}^r_{c,k}$'s are independently sampled from uniform distribution $U(-1, 1)$. The latent variable $X^r_i(t)$'s are generated from Gaussian processes on $[0, 2]$ with the parameters above.

**Setting of $Y^r_j(\cdot)$'s and $Z^r_{ij}(\cdot)$'s.** Furthermore, we generate event sequences for $m$ ($m = 1$, 20 or 100) days. When $m = 1$, the event sequences are generated from the single-level model in (2), which didn't involve the variation $Y_j(t)$ and $Z_{i,j}(t)$. When $m = 20$ or 100, we incorporate $Y_j(t)$ and $Z_{i,j}(t)$ in the intensity function and generate data with the multi-level model in (1).

We further describe the setup of the distributions of $Y^r_j(t)$'s and $Z^r_{i,j}(t)$'s. We let,

$$\tilde{Y}^r_j(t) = \sum_{k=1}^{2} \xi^Y_{r,j,k} \phi^Y_k(t), \quad Z^r_{i,j}(t) = \sum_{k=1}^{4} \xi^Z_{r,i,j,k} \phi^Z_k(t)$$

where $\xi^Y_{r,j,k}$'s and $\xi^Z_{r,i,j,k}$'s are all independent mean-zero normal variables. We set $Var[\xi^Y_{r,j,k}] = 0.2$ and $Var[\xi^Z_{r,i,j,k}] = 0.05$. We set $\{\phi^Y_1(t), \phi^Y_2(t)\} = \{1, \sin(2\pi t)\}$ and $\{\phi^Z_1(t), \phi^Z_2(t), \phi^Z_3(t), \phi^Z_4(t)\} = \{1_{[0,0.5]}, 1_{(0.5,1]}, 1_{(1,1.5]}, 1_{(1.5,2]}\} \times 2\sin(4\pi t)$. Moreover, in order to model the dependence among different days, we let $Y^r_j(t) = 0.8\tilde{Y}^{(}_j t) + 0.6\tilde{Y}^r_{j-1}(t)$ for $j > 1$.

**Evaluation Metric.** For synthetic data, we introduce the criterion *clustering purity* [18] to evaluate the clustering accuracy.

$$\text{Purity} = \frac{1}{n} \sum_{c=1}^{C} \max_{j \in \{1, \cdots, C\}} |\mathcal{W}_c \cap \mathcal{C}_j|,$$

where $\mathcal{W}_c$ is the estimated index set of sequences belonging to the $c$th group, $\mathcal{C}_j$ is the true index set of sequence belonging to the $j$th cluster, and $|\cdot|$ is the cardinality counting the number of elements in a set. The value of *clustering purity* resides in $[0, 1]$ with a higher value indicating a more accurate clustering (=1 if the estimated clusters completely overlap with the truth).

## S.4 Additional Real Data Examples and Details

**Evaluation Metric**  In the real data example, we evaluate and compare clustering stability based on a measure called *clustering consistency* via $K$-trial cross validations [7, 8], as there is no ground truth clustering labels.

It works with the following rationale: because random sampling does not change the clustering structure of data, a clustering method with high consistency should preserve the pairwise relationships of samples in different trials. Specifically, we perform the clustering with $K$ trials. In the $k$-th trial, we randomly separate the accounts into two folds. One fold contains $80\%$ of accounts and serves as the training set, and we predict the cluster memberships of remaining accounts with the trained model. Let $\mathcal{M}_k = \{(i, i')| i, i'$ belong to the same cluster$\}$ enumerate all pairs of accounts with the same cluster index in the $k$-th trial. Then we define the *clustering consistency* as:

$$\text{Clustering Consistency} = \min_{k \in \{1, \cdots, K\}} \sum_{k' \neq k} \sum_{(i,i') \in \mathcal{M}_k} \frac{1\{c^k_i = c^{k'}_{i'}\}}{|K - 1||\mathcal{M}_k|}$$

where $1\{\cdot\}$ is an indicator function and $c^j_i$ denote the learned cluster index of the account $i$ in the $k$-th trial.

**Additional Results on Chase Credit Card Dataset.**  In the credit card transaction dataset, there is a large variation in the frequencies in credit card use across users. We removed the users with fewer than 100 total transactions. The BIC suggests clustering the users into 3 groups. In each cluster, we obtained the estimated surface of covariance function $\Gamma_{x,c}(s, t)$, which is displayed in Figure S1. Compared with clust 2 and 3, the latent process $X_i(t)$ in clust 1 has relatively larger variation. To offer a more straightforward view of the correlation among events, we computed the average correlations as,

$$\overline{\text{Corr}}(r) = \frac{\sum_{|t-s|=r} \widehat{\text{Corr}}(t, s)}{\sum_{|t-s|=r} 1}$$

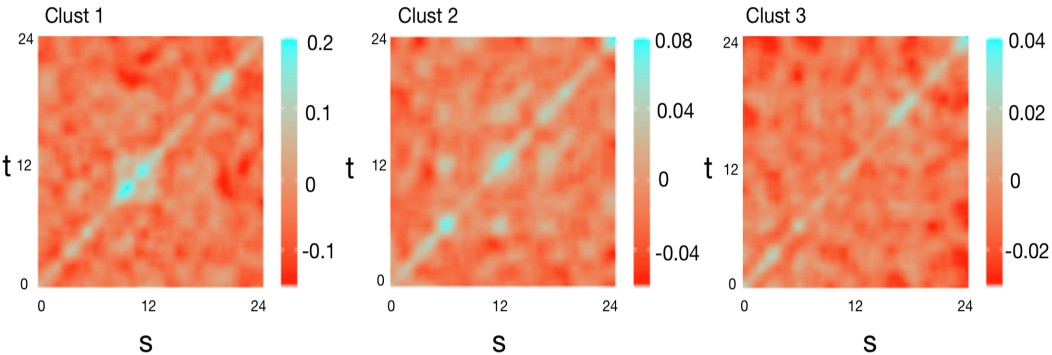

Figure S1: Credit Card Dataset: Estimated $\Gamma_{x,c}(s,t)$ for each cluster;

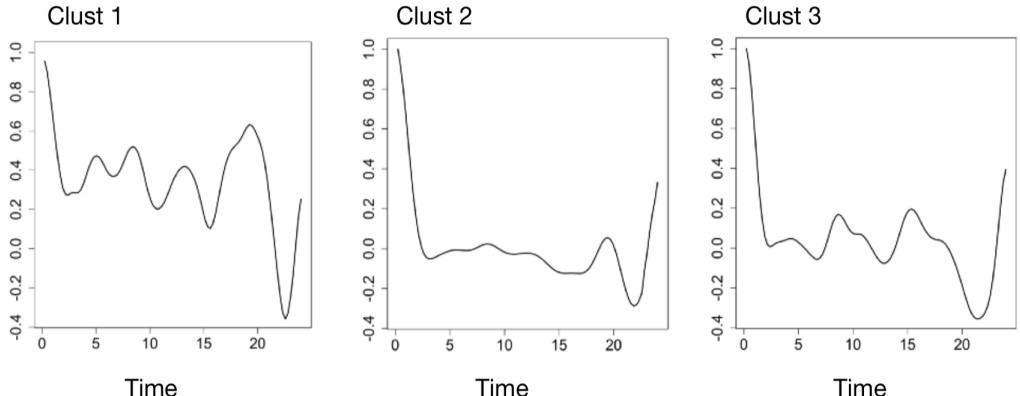

Figure S2: Credit Card Dataset: Averaged correlations versus time lags;

Where $\widehat{\mathrm{Corr}}(t,s) = \widehat{\Gamma}_{x,c}(t,s)/\sqrt{\widehat{\Gamma}_{x,c}(t,t)\widehat{\Gamma}_{x,c}(s,s)}$. Figure S2 displays the averaged correlations versus time lags. There appears to be a periodic pattern in credit card use for clust 1 and 3. The users in clust 1 seemed to use their credit cards most frequently since the plot of clust 1 has the most number of crests. It is consistent with our facts that users in clust 1 averagely used credit cards 3.7 times a day, versus 1.3 times and 2.2 times a day for clust 2 and 3 respectively.