# OpenReview forum: "Row-clustering of a Point Process-valued Matrix"
_NeurIPS.cc/2021/Conference — NeurIPS 2021 Poster_

### Official Review · Reviewer_qLjo · 2021-07-12

**Rating:** 6
**Confidence:** 1

**Summary:**

This paper proposes a mixture model of multi-level marked point processes (MM-MPP), which is inspired by a real-world Twitter dataset. A computationally efficient expectation-solution (ES) algorithm is proposed and shown to outperform two existing methods in numerical experiments.

**Main Review:**

This work extends the existing MM-MPP to a mixture model setting, accounting for heterogeneity in data. This is largely inspired by a Twitter dataset, though it is potentially suitable for many other applications as well.

Pro: the overall structure of this paper is clean and easy to follow. The models (Sec 3) and algorithms (Sec 4), as well as settings and results of numerical experiments (Sec 5), are presented and explained in detail. In particular, the proposed ES algorithm admits close-form solution in its intermediate steps, which allows efficient computation.

Con: the proposed ES algorithm doesn't come with any theoretical guarantees, and one of my concern is about its convergence. It is well known that typically EM-type algorithm can get stuck at local minimums; the current paper has no discussion on this issue. I would suggest checking (at least) empirically whether the proposed algorithm is indeed robust/insensitive to random initialization, e.g. with 100 random initializations, how many times does it converge to a good solution?

A minor suggestion: in Tables 1 and 2, it would be better to include not just a single number, but also a confidence interval, to account for uncertainty/randomness in data and/or algorithms. It is also helpful (and standard) to use bold font to highlight the best performance under each setting.

Typo: Line 7, "Exponential-Solution" -> "Expectation-Solution".

**Time Spent Reviewing:**

2

---

> ### Author Response · Authors · 2021-08-10
> **Response to Reviewer qLjo**
>
> * **Additional numerical examples:** We thank the reviewer's careful review of our manuscript. We agree with the reviewer's comment that: ``This is largely inspired by a Twitter dataset, though it is potentially suitable for many other applications as well."  In the revised manuscript,  we plan to add two additional real data examples to illustrate our methods.
> We have added two additional real data examples to illustrate our methods in the revised manuscript.
>     * Chase Credit Card Dataset: We applied the proposed method to a credit card transaction data set, which was collected by Worldline and the Machine Learning Group at Université Libre de Bruxelles and can be accessed via kaggle database. The data set contains 641,914 transaction records of 5,000 European credit card customers duringthe period covering January 1 to December 31, 2016.
> We treated the one-day transaction records of each account as an event sequences and applied the multi-level LGCP model to the dataset. For comparisons, we also applied the single-level model to the data by concatenating the event sequences of each account into a single long sequence. The single-level model clustered the accounts into 5 groups. The results seemed to cluster card users mainly by their overall frequency of transactions while incapable of capturing their daily patterns of transactions.
> For the multi-level model, we obtained 6 clusters under the BIC criteria. Two heterogeneous daily transaction patterns can be recognized although they have the similar overall event frequency. We further estimate the $\hat{Y}_j^r$ using the method offered in Xu et al (2020) and we found the notable monthly periodic property of $\hat{Y}_j^r$.
>     * New York City Taxi Dataset: The NYC taxi dataset contains around 173 million trip records of individual Taxi from Jan. 1 to Dec 31, 2013.  Each record includes the time and locations (in the form of latitude/longitude coordinates) of pick-up and drop-off passengers. We formed the temporal event sequences using the pick-up time. Also we mapped the pick-up location into 51 Council districts in NYC as the markers of events.
> We applied the multi-level marked LGCP to the dataset, and here we summarize some preliminary findings. Overall, our method showed a higher clustering consistency than of discrete Frechet distance-based method (DF, Pei, 2003) and Dirichlet mixture of Hawkes processes (DMHP, Xu, 2017). Our method produced 10 clusters, whereas DMHP produced 14 clusters and DF produced 23 clusters. We observed that DMHP seems to cluster taxis by their overall pick-up frequency and time intervals between events. In contrast, our method is capable of clustering taxis taking into account their daily schedule patterns and  boroughs.
>
> * **Convergence**
>
> Elashoff and Ryan (2004) established a general theory for the local convergence of the E-S algorithm. Under two regularity conditions on the first derivative of estimating equations, there exists a neighborhood of solution such that the algorithm has a guaranteed convergence if initializing the unknown parameter with a value within this neighborhood. The theory is proved by treating ES as a Block Newton-Gauss-Seidel algorithm and then directly applying the convergence results in Ortega (1972, p. 147). Our ES algorithm is a special case of Elashoff and Ryan (2004), so their general theories apply to our case.
> We will add some of the above discussions in the revised manuscript to clarify the motivations and rationals of the E-S algorithm.
> We have also taken the reviewer's suggestion and numerically checked the convergence of our algorithm. We consider the same setting as the synthetic data example in section 5.1 and Appendix S.3 with $C=3$, $R=5$, $n=500$, $m=20$. We initialize the algorithm by first assigning random cluster memberships to subjects and then calculating the initial value of parameters within each cluster. The value of the log-likelihood becomes stable after approximately 20 iterations and the convergence of parameters will be achieved after around 40 iterations.
>
> However, if we kick off with the initial parameters which were obtained by quick distance-based methods like Pei et al (2013), then the parameters will converge around 10 iterations.
>
> * **Minor comments**
> We thank the reviewer for this suggestion. In the revised manuscript, we will highlight the results of the best-performing model and also report the standard errors of our evaluation metrics based on 100 repeated experiments.  We thank the reviewer for pointing out the typo. We have corrected it.
>
>
> * **References**
>
>     * Michael Elashoff and Louise Ryan. An EM algorithm for estimating equations. Journal of  Computational and Graphical Statistics, 13(1):48–65, 2004.
>     * Ortega, J. M.  Numerical Analysis: a Second Course. New York: Academic Press, 1972.
>     * Tao Pei, Xi Gong, Shih-Lung Shaw, Ting Ma, and Chenghu Zhou. Clustering of temporal event processes. International Journal of Geographical Information Science, 2013.
>     * Ganggang Xu, Ming Wang, Jiangze Bian, Hui Huang, Timothy Burch, Sandro Andrade, Jingfei Zhang, and Yongtao Guan. Semi-parametric learning of structured temporal point processes. Journal of machine learning research, 2020.
>     * Hongteng Xu and Hongyuan Zha. A dirichlet mixture model of hawkes processes for event sequence clustering. NeurIPS, 2017.

---

> > ### Comment · Reviewer_qLjo · 2021-08-19
> > **After rebuttal**
> >
> > Thanks for the authors' detailed responses. I'd like to raise my score from 5 to 6 albeit my low confidence in my assessment.

---

> > > ### Author Response · Authors · 2021-08-20
> > > **Additional Response to Reviewer qLjo**
> > >
> > >
> > > We offer our sincere gratitude to your response. I'd like to report more details.
> > >
> > > * **Convergence**
> > > We initialize the algorithm by first assigning random cluster memberships to subjects and then calculating the initial value of parameters within each cluster. With the setting that $R=5$, $C=2$, $n=500$, we experimented $100$ heterogeneous simulated datasets and tried $100$ random initial values for each dataset. Consequently, estimates from $92.7$% of attempts converged in the proximity of the true parameters when $m=1$. However, almost $100$% would converge to the proximity of the true parameters when $m=20$ and $m=100$, since the difference among individuals will be augmented with $m$ increasing.
> > >
> > > * **Computation**
> > > The proposed ES framework boasts of its computational efficiency in comparison with the EM algorithm. We will share the code and report the computation times of non-GPU-based E-M, non-GPU-based E-S, and GPU-based E-S algorithms in the revised manuscript.  Here, we report part of run-time (in seconds) of 20 iterations in the case that $R=5$, $C=3$, $m=20$ and size of sampling $=10,000$.
> > >
> > > |  | GPU-ES (RTX 8000 48G GPU) | CPU-ES (i7-7700HQ CPU) | CPU-EM (i7-7700HQ CPU) |
> > > |:----:| :----: | :----: | :----: |
> > > | n=500 | 30.09 | 275.87 | 568.36 |
> > > | n=1000 | 51.42 | 505.07 | 1105.46

---

### Official Review · Reviewer_3ktE · 2021-07-16

**Rating:** 6
**Confidence:** 4

**Summary:**

This paper studies an interesting problem, which aims to identify potential heterogeneity in the observed data where structured event data are harvested from various platforms. The authors imposes a matrix structure to repeatedly observed marked point processes, and propose a mixture model of multi-level marked point processes. In particular, the data is structured into a matrix, where each entry contains a sequence of events; these events are modeled by marked log-Gaussian Cox processes; the ultimate goal is to cluster rows of such a matrix. They also propose an efficient semi-parametric Exponential-Solution (ES) algorithm combined with functional principal component analysis of point processes for model estimation.

**Limitations And Societal Impact:**

- I don't quite understand how the event data can be divided by days? Would this separation cause discontinuity to the original sequence? And how to evaluate the influence of such data preprocessing to the final results?
- Since the entire paper is based on a key assumption that event sequences from each cluster are realizations of a multi-level log-Gaussian Cox process, can the authors add more explanation why this assumption applies to the most of the applications?
- The choices of the bandwidth is significant to the performance of the proposed model. The author mentioned the bandwidth is determined adaptively at each iteration via MLE. It seems to be expensive to carry out in practice.

**Main Review:**

The paper is clearly written and easy to follow. The main contribution of this paper is introducing the ES algorithm, which is similar to EM algorithm, aiming to facilitate the computation in estimating the latent variables by adopting the kernel method.

**Time Spent Reviewing:**

2

---

> ### Author Response · Authors · 2021-08-10
> **Author response to Reviewer 3KtE**
>
> We thank the reviewer for the constructive comments.
>
> * **Segmentation by days:**
>  We focus on repeatedly observed marked point processes in this manuscript and use daily event sequences as an illustration. The key assumption behind such segmentation is that the user posting behavior is partly determined by user-specific characteristics ($X_i(t)$) that do not change from day to day. The multi-level model further incorporates contributions from day-related factors $Y_j(t)$ (such as weekday effect, weekend effect, or holiday effect) to the posting behavior. However, the purpose of the clustering is to cluster users based on user-related characteristics ($X_i(t)$), which is not lost by aggregation even though there may be discontinuity to the original sequence. The impact of such data prepossessing will be on the estimation of $Y_j(t)$, but not $X_i(t)$ since we assume that $X_i(t)$'s and $Y_j(t)$'s are independent.  In this sense, the proposed multi-level model allows us to borrow and learn shared patterns ($X_i(t)$'s) across days while accounting for day-to-day variations and handling discontinuities in the observation time window.  Finally, we would like to comment that the proposed method also applies to other repeated measurements of point processes such as signals on waves and specific segments on genes. Many real data, especially those involving human activities, are naturally repeatedly observed by days and have none or inactive events during certain time periods in a day. Examples include the twitter data we considered in our analysis, credit card and stock trading transactions, to name a few.
>
> * **Would this separation cause discontinuity to the original sequence?**
> In many applications such as the social media posting example, discontinuity naturally exists when no activity occurs for an extended period of time. This is what motivates us to study an individual's daily activity patterns, which naturally have some similarities for the same user on different days. The introduction of the day-level latent process $Y_j(t)$'s further allows for dependence between repeated segments. We believe this model can be used to capture additional aspects of the observed data compared to the original sequence, as we will demonstrate through the following experiments.
>
> * **How to evaluate the influence of such data preprocessing on the final results?**
> We have conducted some additional experiments, including synthetic data analysis and real-world data analysis, to be reported in a future revision. In these experiments, we compare the proposed multi-level model to the single-level model (MM-SPP) applying to the original sequence. In the synthetic data analysis, the table below reports the Clustering Purity in the case with $n=500$, $C=2$, $R=5$, where we can see that the Clustering Purity dropped dramatically while applying the MM-SPP to the original long sequence.
>
> |  | m=5 | m=20 | m=100 |
> |:----:| :----: | :----: | :----: |
> | Multi-Level (Repeated Segments) | 0.862 | 0.947 | 0.988 |
> | Single-Level (Original Sequence) | 0.815 | 0.743 | 0.723 |
>
>    Besides, we have now conducted similar experiments in the real-world dataset. When studying the Chase Credit Card Dataset, the single-level model based on the original sequence yields 5 clusters using the BIC criteria. This method tends to cluster card users mainly by their overall frequency of transactions while incapable of capturing their intra-day transaction patterns. Using the multi-level model based on repeated segments, we obtained 6 clusters. Two heterogeneous daily transaction patterns can be recognized although they have similar overall event frequencies. We further estimate the $\hat{Y}_j^r$ using the method offered in Xu et al (2020) and we found the notable monthly periodic property of $\hat{Y}_j^r$.
>
> For the New York City Taxi Dataset, using the multi-level model based on repeated segments, we are able to identify two groups who have similar overall pick-up frequencies but have heterogeneous daily working schedules. Specifically, one group is more likely to work in the daytime and another group always work at night. From all those experiments, we think the multi-level model can more accurately describe an individual's daily activity behavior and daily schedule.
>
> * **Model assumptions:** We have no intention to claim that the multi-level LGCP applies to most of the applications, but there are increasingly more applications that may be more suitable to use this model. For human activity data such as social media posting and stock transaction activities, it is reasonable to believe that, in addition to user-specific characteristics, many external factors also contribute to the activity patterns. For example, spatial locations of the user or overall macroeconomic environment.  Existing models such as multi-variate Hawkes models rely heavily on interval user factors by assuming that human activities are self-exciting, which may be overly simplistic. Another appealing property of the multi-level LGCP model is that it can be easily extended to include more than two factors into the model and hence has the potential to handle more complicated point pattern data. In principle, the multi-level model is essentially an extension of the classical (two-way) analysis of variance (ANOVA) model to the case where each observed point in ANOVA is replaced by a marked point process. One can design any reasonable experiments using various factors to obtain replicated marked-point processes (in a fixed time domain, such as a day). Taking the stock trading data studied in Xu et al. (2020) for example, we can consider user stock trading activities on the same day but at different trading locations. In this case, the first level will be the user-related characteristics and the second level will be geographical locations. Such a data set may reflect how user-related characteristics and spatial economic factors in a country contribute to the user stock trading activities. A three-level model (user, day, and spatial location) can also be considered in the same fashion as in Xu et al. (2020). We refer readers to Xu et al (2020) for a more comprehensive discussion on the advantages of the multi-level LGCP model.
>
> * **Choice of bandwidth:**
> We thank the reviewer for this helpful suggestion. To reduce the computation with adaptive bandwidths,  we pre-computed the two terms $a_i^{r,r'}(s,t;h)$ and $b_i^r(t;h)$ that involve kernel functions (see the top of page 5) for each candidate bandwidth. At each iteration, (12) and (13) in S-step were linear combinations of the pr-calculated components. Moreover, our E-step does not involve kernel functions. Therefore, the computation was affordable when we used adaptive bandwidths selections. But we will take the reviewer's suggestion to experiment with the method that only adaptively determines bandwidth in the first few iterations of E-S and fix the bandwidth for the remaining iterations.
>
>    Also, the proposed ES framework boasts of its computational efficiency in comparison with the EM algorithm. We will share the code and report the computation times of non-GPU-based E-M, non-GPU-based E-S, and GPU-based E-S algorithms in the revised manuscript.  Here, we report part of run-time (in seconds) of 20 iterations in the case that $R=5$, $C=3$, $m=20$ and size of sampling $=10,000$.
>
> |  | GPU-ES (RTX 8000 48G GPU) | CPU-ES (i7-7700HQ CPU) | CPU-EM (i7-7700HQ CPU) |
> |:----:| :----: | :----: | :----: |
> | n=500 | 30.09 | 275.87 | 568.36 |
> | n=1000 | 51.42 | 505.07 | 1105.46
>
> * **References**
>
>    * Ganggang Xu, Ming Wang, Jiangze Bian, Hui Huang, Timothy Burch, Sandro Andrade, Jingfei Zhang, and Yongtao Guan. Semi-parametric learning of structured temporal point processes. Journal of machine learning research, 2020.

---

> > ### Comment · Reviewer_3ktE · 2021-08-22
> > **Reply to author's response**
> >
> > Thank you for the detailed comments. The responses have address most of my concern. I will raise my score to 6.

---

### Official Review · Reviewer_J9LW · 2021-07-19

**Rating:** 7
**Confidence:** 2

**Summary:**

This paper studies clustering based on multi-dimensional data array that can be represented by a point process, which is motivated by Twitter activity data. It proposes a mixture model, where each mixture is a multivariate Gaussian process plus noise and addition structure. In order to identify the mixtures and the components, this paper uses the E-S algorithm, together with several approximation and sampling tricks. Experiments on the Twitter activity data show that the universities under study can be grouped into three clusters, which has meaningful interpretations.

**Limitations And Societal Impact:**

Not applicable.

**Main Review:**

This paper seems to provide a useful statistical methodology for a complex heterogeneous dataset. While I do not claim expertise in this field, this paper does seem to present a practical E-S algorithm for statistical analysis. One drawback of this paper is that the motivating data example looks quite limiting, and that the data example does not reveal interesting discoveries beyond obvious facts (universities can be grouped according to their tiers, etc). I am not criticizing but I feel that a better data example would lead to a much wider appeal to the general audience.

**Time Spent Reviewing:**

3

---

> ### Author Response · Authors · 2021-08-10
> **Response to Reviewer J9LW**
>
> We have added two additional real data examples to illustrate our methods in the revised manuscript.
>
> * **Chase Credit Card Dataset:** We applied the proposed method to a credit card transaction data set, which was collected by Worldline and the Machine Learning Group at Université Libre de Bruxelles and can be accessed via kaggle database. The data set contains 641,914 transaction records of 5,000 European credit card customers duringthe period covering January 1 to December 31, 2016.
>
>  * **New York City Taxi Dataset**: The NYC taxi dataset contains around 173 million trip records of individual Taxi from Jan. 1 to Dec 31, 2013.  Each record includes the time and locations (in the form of latitude/longitude coordinates) of pick-up and drop-off passengers. We formed the temporal event sequences using the pick-up time. Also we mapped the pick-up location into 51 Council districts in NYC as the markers of events.
>
> Main Findings:
>
> * **Chase Credit Card Dataset:** We treated the one-day transaction records of each account as an event sequences and applied the multi-level LGCP model to the dataset. For comparisons, we also applied the single-level model to the data by concatenating the event sequences of each account into a single long sequence. The single-level model clustered the accounts into 5 groups. The results seemed to cluster card users mainly by their overall frequency of transactions while incapable of capturing their daily patterns of transactions.
> For the multi-level model, we obtained 6 clusters under the BIC criteria. Two heterogeneous daily transaction patterns can be recognized although they have the similar overall event frequency. We further estimate the $\hat{Y}_j^r$ using the method offered in Xu et al (2020) and we found the notable monthly periodic property of $\hat{Y}_j^r$.
>
> * **New York City Taxi Dataset**:
> We applied the multi-level marked LGCP to the dataset, and here we summarize some preliminary findings. Overall, our method showed a higher clustering consistency than of discrete Frechet distance-based method (DF, Pei, 2003) and Dirichlet mixture of Hawkes processes (DMHP, Xu, 2017). Our method produced 10 clusters, whereas DMHP produced 14 clusters and DF produced 23 clusters. We observed that DMHP seems to cluster taxis by their overall pick-up frequency and time intervals between events. In contrast, our method is capable of clustering taxis taking into account their daily schedule patterns and  boroughs.
>
> * **References**
>
>    * Tao Pei, Xi Gong, Shih-Lung Shaw, Ting Ma, and Chenghu Zhou. Clustering of temporal event processes. International Journal of Geographical Information Science, 2013.
>    * Ganggang Xu, Ming Wang, Jiangze Bian, Hui Huang, Timothy Burch, Sandro Andrade, Jingfei Zhang, and Yongtao Guan. Semi-parametric learning of structured temporal point processes. Journal of machine learning research, 2020.
>    * Hongteng Xu and Hongyuan Zha. A dirichlet mixture model of hawkes processes for event sequence clustering. NeurIPS, 2017.

---

### Official Review · Reviewer_DpLt · 2021-07-19

**Rating:** 6
**Confidence:** 3

**Summary:**

The problem is equivalent to a marked point process, i.e. a set of times with associated categorical marks. In the main application of Twitter data modelling, the times represent when a tweet was made, and the marks are the cartesian product of twitter handle and tweet type (tweet, retweet, reply).

The data is split by day and by twitter handle, yielding a 2D matrix of marked point processes with the time (without date) and tweet type for each element of the matrix (so each element of this matrix is a sequence of random length). A heirarchical model is fit to this data, where the intensity for any element of this matrix is a linear combination of that for the day, for the twitter handle, and a third term for the interaction between both, with latent GP for each such term. So far this model comes mainly from [26]. Then, a cluster assignment variable omega is introduced which selects from multiple GP models for the twitter handle component of the intensity function, thereby yielding a mixture model with mixture parameters being GP distributed and capturing the intensity of tweets throughout the day, for different clusters of users.

Inference is performed with a form of EM, namely ES or Expectation Solution. This is rather non-trivial and involves significant analytical and numerical adeptness to execute. The paper is therefore technically impressive, and demonstrates excellent technical prowess in probabilistic inference.

**Limitations And Societal Impact:**

Yes.

**Main Review:**

The paper is interesting and the model seems interesting for an ML audience, given the rise of social media and associated data.

The main contribution is in the form of a non-trivial inference scheme. Given the depth of numerical challenges here, it is hard to evaluate the paper. It would have been nice to have received code. At the very least, failing providing the code, it is essential that the experimental setup be reproducible so that subsequent authors can compare. Also, it is a slight disappointment that the paper includes just one significant real-world  numerical evaluation, on a dataset which is somewhat arbitrary in construction and evaluation. However, it remains impressive that the authors could put together this non-trivial scheme and ostensibly make it work.

Some literature is missing. On line 40: "most existing work rely on strong parametric assumptions" is questionable. Much recent work uses latent GP intensities for Poisson models and also for latent GP Hawkes' process triggerring kernels. Please cite these works and remove this misleading statement.

The model is fun but has some strong assumptions, chiefly it is US centric given the segmentation by "day".

The title is a little obfuscated and might make the paper easy to miss.


**Time Spent Reviewing:**

2

---

> ### Author Response · Authors · 2021-08-10
> **Response to Reviewer Dplt**
>
> We thank the reviewer for carefully reviewing our manuscript and providing many constructive comments
>
> * **Reproducibility:** We completely agree with the reviewer that reproducibility is an important component of research.  We will include the Github link to our code and data in the revised manuscript.
>
> * **Additional data examples**: we plan to add two additional real data examples to illustrate our methods.
>    * Chase Credit Card Dataset: We applied the proposed method to a credit card transaction data set, which was collected by Worldline and the Machine Learning Group at Université Libre de Bruxelles and can be accessed via kaggle database. The data set contains 641,914 transaction records of 5,000 European credit card customers duringthe period covering January 1 to December 31, 2016.
> We treated the one-day transaction records of each account as an event sequences and applied the multi-level LGCP model to the dataset. For comparisons, we also applied the single-level model to the data by concatenating the event sequences of each account into a single long sequence. The single-level model clustered the accounts into 5 groups. The results seemed to cluster card users mainly by their overall frequency of transactions while incapable of capturing their daily patterns of transactions.
> For the multi-level model, we obtained 6 clusters under the BIC criteria. Two heterogeneous daily transaction patterns can be recognized although they have the similar overall event frequency. We further estimate the $\hat{Y}_j^r$ using the method offered in Xu et al (2020) and we found the notable monthly periodic property of $\hat{Y}_j^r$.
>     * New York City Taxi Dataset: The NYC taxi dataset contains around 173 million trip records of individual Taxi from Jan. 1 to Dec 31, 2013.  Each record includes the time and locations (in the form of latitude/longitude coordinates) of pick-up and drop-off passengers. We formed the temporal event sequences using the pick-up time. Also we mapped the pick-up location into 51 Council districts in NYC as the markers of events.
> We applied the multi-level marked LGCP to the dataset, and here we summarize some preliminary findings. Overall, our method showed a higher clustering consistency than of discrete Frechet distance-based method (DF, Pei, 2003) and Dirichlet mixture of Hawkes processes (DMHP, Xu, 2017). Our method produced 10 clusters, whereas DMHP produced 14 clusters and DF produced 23 clusters. We observed that DMHP seems to cluster taxis by their overall pick-up frequency and time intervals between events. In contrast, our method is capable of clustering taxis taking into account their daily schedule patterns and  boroughs.
>
> * **Clarification on literature review:** ``Parametric assumptions" here are referring to the use of parametric covariance functions in latent GP models. We did not impose such parametric assumptions in our LGCP model and instead use functional PCA to estimate covariance from data. We will clarify this point and cite relevant literature in the revision.
>
> * **Segmentation by days:** The segmentation by  ``day" is merely due to the characteristic of the motivating Twitter data set. Many other factors can be used when creating replicated mark-point processes. Taking the stock trading data studied in Xu et al. (2020) for example, we can consider user stock trading activities on the same day but at different trading locations. In this case, the first level will be the user-related characteristics and the second level will be geographical locations. Such a data set may reflect how user-related characteristics and spatial economic factors in a country contribute to the user stock trading activities. In principle, the multi-level model is essentially an extension of the classical (two-way) analysis of variance (ANOVA) model to the case where each observed point in ANOVA is a marked point process. One can design any reasonable experiments using various factors to obtain replicated marked-point process (in a fixed time domain, such as a day).
>
> * **References**
>
>    * Tao Pei, Xi Gong, Shih-Lung Shaw, Ting Ma, and Chenghu Zhou. Clustering of temporal event processes. International Journal of Geographical Information Science, 2013.
>    * Ganggang Xu, Ming Wang, Jiangze Bian, Hui Huang, Timothy Burch, Sandro Andrade, Jingfei Zhang, and Yongtao Guan. Semi-parametric learning of structured temporal point processes. Journal of machine learning research, 2020.
>    * Hongteng Xu and Hongyuan Zha. A dirichlet mixture model of hawkes processes for event sequence clustering. NeurIPS, 2017.

---

### Decision · Program_Chairs · 2021-09-28

**Decision:**

Accept (Poster)

**Comment:**


The paper studies structured point process data. The model fitting is achieved using the ES algorithm (an extension of the EM algorithm). The problem considered is interesting and relevant to the machine learning community. The benefit of leveraging the structure for point process models is demonstrated using real data. However, it may be challenging to develop a performance guarantee for such a type of algorithm. So the contribution is maybe in developing a structure-aware model and algorithms for model fitting. The paper addressed the computational efficiency of the algorithm.

**Consistency Experiment:**

NeurIPS has a long history of experimentation. In 2014, NeurIPS ran an experiment in which 10% of submissions were reviewed by two independent committees to quantify the randomness in the review process. This year, we repeated a variant of this experiment to see how the quality of the review process has changed over time.  This paper was part of the experiment and was therefore assigned to two committees (consisting of reviewers, an Area Chair, and a Senior Area Chair) that reached independent decisions.  If both committees made the same recommendation, this recommendation was followed. If a single committee recommended acceptance, the paper was accepted (with the exception of a few cases in which the other committee identified what we considered a fatal flaw, e.g., an error in a key result).

This copy’s committee reached the following decision: **Accept (Poster)**

The other committee assigned to the paper recommended **Reject**.  You can find the other set of reviews, along with any follow up discussion with the authors here:
https://openreview.net/forum?id=YXy_2b5wufe